# Synthesis and superconductivity in yttrium-cerium hydrides at high pressures

Liu-Cheng Chen [1,2], Tao Luo[1,2], Zi-Yu Cao [2,3], Philip Dalladay-Simpson [2], Ge Huang[2], Di Peng [2], Li-Li Zhang[4], Federico Aiace Gorelli[2,5], Guo-Hua Zhong [6,7], Hai-Qing Lin[8] & Xiao-Jia Chen [9] ✉

Further increasing the critical temperature and/or decreasing the stabilized pressure are the general hopes for the hydride superconductors. Inspired by the low stabilized pressure associated with Ce $4f$ electrons in superconducting cerium superhydride and the high critical temperature in yttrium super-hydride, we carry out seven independent runs to synthesize yttrium-cerium alloy hydrides. The synthetic process is examined by the Raman scattering and X-ray diffraction measurements. The superconductivity is obtained from the observed zero-resistance state with the detected onset critical temperatures in the range of 97-141 K. The upper critical field towards 0 K at pressure of 124 GPa is determined to be between 56 and 78 T by extrapolation of the results of the electrical transport measurements at applied magnetic fields. The analysis of the structural data and theoretical calculations suggest that the phase of $Y_{0.5}Ce_{0.5}H_9$ in hexagonal structure with the space group of $P6_3/mmc$ is stable in the studied pressure range. These results indicate that alloying superhydrides indeed can maintain relatively high critical temperature at relatively modest pressures accessible by laboratory conditions.

Rare earth hydrides have been found to exhibit near room temperature superconductivity benefiting from the chemical pre-compression induced by the interaction between hydrogen and tetragen atoms[1,2]. A typical example among them is the experimental discoveries of super-conductivity at the critical temperature $T_c$ as high as 250–260 K at pressure of ~170 GPa[3,4] for clathrate $LaH_{10}$ with $Fm\bar{3}m$ structure[4,5] based on the early theoretical predictions[6,7]. In parallel, yttrium hydrides as another attractive rare earth hydride system have drawn a lot of attentions from the predictions[6–8] and experimental realization[9–12] of superconductivity. The experiments reported the $I4/mmm$-$YH_4$ phase with the maximum $T_c$ of 88 K at 155 GPa[9], the $Im\bar{3}m$-$YH_6$ phase with a similar $T_c$ near 220 K in the pressure range of 166–180 GPa[10,11], and the

$P6_3/mmc$-$YH_9$ phase with the maximum $T_c$ ~ 243 K at 201 GPa[10] or with the significantly high maximum $T_c$ ~ 262 K at 182 GPa[12]. It is clear that the experimentally obtained $YH_x$ superconductors are only stable at extreme pressures (above ~155 GPa)[9–12]. Meanwhile, theoretical[6,13–15] and experimental[13,14] efforts established that cerium superhydrides can be stabilized at low pressures below 100 GPa but can be superconductive as well. The enhanced chemical pre-compression in $CeH_9$ was sug-gested to be associated with the delocalized nature of Ce $4f$ electrons[16]. Recently, cerium superhydrides were found to exhibit super-conductivity with the maximum $T_c$ of 115 K for $Fm\bar{3}m$-$CeH_{10}$ at 95 GPa and $T_c$ of 57 K for $P6_3/mmc$-$CeH_9$ at 88 GPa[17]. This finding was confirmed for $CeH_{10}$ with even higher $T_c$ values at the corresponding modest

[1]School of Science, Harbin Institute of Technology, Shenzhen 518055, China. [2]Center for High Pressure Science and Technology Advanced Research, Shanghai 201203, China. [3]Center for Quantum Materials and Superconductivity (CQMS) and Department of Physics, Sungkyunkwan University, Suwon 16419, Republic of Korea. [4]Shanghai Synchrotron Radiation Facility, Shanghai Advanced Research Institute, Chinese Academy of Sciences, Shanghai 201204, China. [5]National Institute of Optics (INO-CNR) and European Laboratory for Non-Linear Spectroscopy (LENS), Via N. Carrara 1, 50019 Sesto Fiorentino (Florence), Italy. [6]Shenzhen Institute of Advanced Technology, Chinese Academy of Sciences, Shenzhen 518055, China. [7]University of Chinese Academy of Sciences, Beijing 100049, China. [8]School of Physics, Zhejiang University, Hangzhou 310058, China. [9]Department of Physics and Texas Center for Superconductivity, University of Houston, Houston, TX 77204, USA. ✉e-mail: xjchen@uh.edu

pressures[18]. Meanwhile, the superconductivity in this superconductor has been found to follow the Bardeen-Cooper-Schrieffer theory with a moderate coupling strength from the determination of its superconducting gap[18]. Unlike La-H and Y-H systems, the stabilized pressure is dramatically reduced in the Ce-H compounds.

The exploration of high-$T_c$ superconductivity in superhydrides at low pressures or even ambient pressure is highly demanded. Compared with binary hydrides, ternary alloy hydrides possess diverse chemical compositions and thus provide more abundant structures for the operation with the advantages of different elements[19–22]. Interestingly, a series of lanthanum-yttrium ternary hydrides was reported to possess superconductivity with the maximum $T_c$ of 253 K at pressures of 170–196 GPa[23], indicating that hydrides can be stabilized in solid solutions at relatively low pressures. Superconductivity in lanthanum-cerium ternary superhydrides with $T_c$ ~ 176 K was strikingly preserved to ~100 GPa[24,25]. Theoretical calculations demonstrated that a series of ternary superhydrides can hold high $T_c'$ s at relatively low stabilized pressures (~50 GPa)[19,21]. Therefore, exploring superconductivity in ternary hydrides towards high $T_c$ at modest pressures from the experimental side is an interesting direction. For such a purpose, we choose $Y_{0.5}Ce_{0.5}$ alloy and synthesize such alloy hydrides with the combined advantages of the low synthesized pressure in Ce-H and high $T_c$ in Y-H compounds.

## Results and discussions

### Synthetic process of $Y_{0.5}Ce_{0.5}$ hydrides

The $Y_{0.5}Ce_{0.5}$ pieces with a thickness of about 1–2 $\mu m$ were loaded into the sample chamber with ammonia borane $NH_3BH_3$ (AB) as the pressure transmitting medium and hydrogen source (Fig. 1a). The samples were heated around 2000 K from the direction of AB side by an YAG laser after compressing to a desirable pressure value (Supplementary Table 1). After laser heating, all the samples (Cell-1 to Cell-7) change dramatically in their shapes and colors (Fig. 1b and Supplementary Fig. 1). Raman spectroscopy is used to detect the vibrational properties of various phases and to identify the formation of phase after the chemical reaction. To eliminate the scattering signals from AB, $c$-BN,

and PtH, the direction of the exciting laser for the Raman scattering is from the alloy side (Top panel of Fig. 1a). The Raman vibron ($v_1$) of $H_2$ is detected with large intensity for the sample after heating, indicating the presence of hydrogen in favor of the formation of $Y_{0.5}Ce_{0.5}$ hydrides (Fig. 1c and Supplementary Fig. 2). The scaled Raman spectra at low frequencies (~300–1000 cm$^{-1}$) display a hump, which consists of the rotational bands of $H_2$, named S(0) and S(1), and the optical phonon associated to the transverse optical $E_{2g}$ mode[26,27]. The Raman bands marked by asterisks at lower frequencies should be the phonon modes from the synthesized alloy hydrides, because of the absence of any Raman active modes for $H_2$ in this region at the studied pressure[26,27]. Thus, we can conclude that the chemical reaction takes place in the sample chamber judging from the micrographs and Raman spectra (Fig. 1 and Supplementary Figs. 1 and 2).

### Superconductivity under pressure

To probe superconductivity in the synthesized $Y_{0.5}Ce_{0.5}$ alloy hydrides, we performed the electrical transport measurements under pressure. Representative temperature-dependent resistance data at high pressures are shown in Fig. 2a. Superconducting transitions can be clearly seen, as evidenced by the sharp drop of the resistance occurring at 129, 113, and 136 K at pressure of about 114, 195, 124 GPa for Cell-3, Cell-6, and Cell-7, respectively. In the resistance measurements, the zero-resistance state was realized for the samples in Cell-3, Cell-6, and Cell-7. In Supplementary Fig. 3, we showed the results of the temperature-dependent resistance measurements on $Y_{0.5}Ce_{0.5}$ alloy before heating and $Y_{0.5}Ce_{0.5}$ hydrides after heating in Cell-6. The large $T_c$ differences before and after heating also indicate the high-$T_c$ superconductivity is from the synthesized alloy hydrides. By putting the determined $T_c$ values together from the temperature-dependent resistance measurements (Fig. 2a and Supplementary Figs. 3–8), we obtain the systematic evolution of $T_c$ with pressure of synthesized $Y_{0.5}Ce_{0.5}$ hydrides (Fig. 2b). With the increase of pressure, $T_c$ is found to increase with pressure and then slightly decrease after passing the maximum of 141 K at an optimal pressure of around 130 GPa.

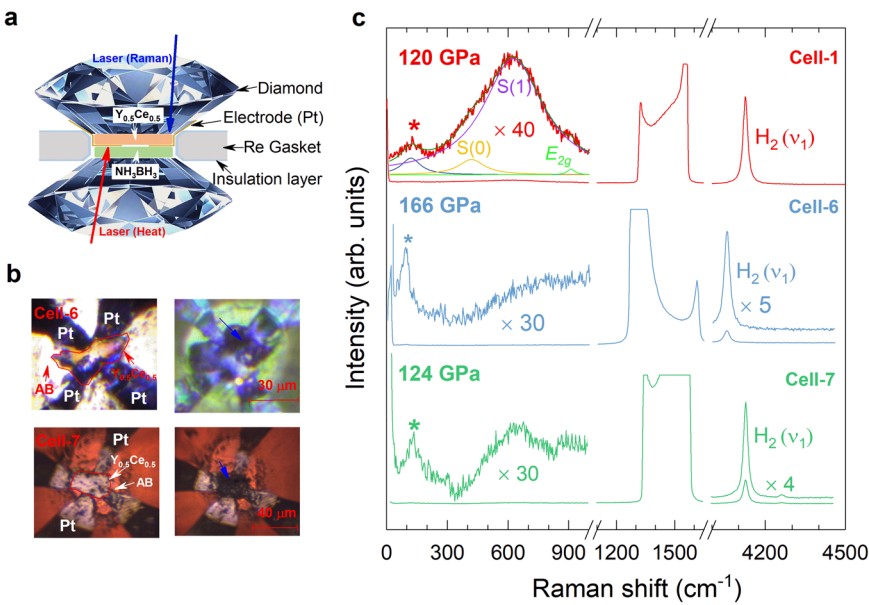

**Fig. 1 | Synthesis of $Y_{0.5}Ce_{0.5}$ hydrides at extreme conditions (high pressure and high temperature). a** Schematic diagram of the experimental setup for the measurements. The arrows represent the laser beam directions for the Raman scattering and heating measurements. **b** Optical micrographs of the sample chambers containing $NH_3BH_3$ (AB) and Pt electrodes in the representative cells (Cell-6 and Cell-7) before and after laser heating. The edges of Y-Ce film are marked with the red lines, and the blue arrows in the right photos indicate the parts with apparent changes after heating. **c** Raman spectra for the synthesized $Y_{0.5}Ce_{0.5}$ hydrides collected at the apparent-changing parts (blue arrows in the right Fig. 1b) in the sample chamber. The Raman bands of the diamond and $H_2$ after laser heating are presented. The low-frequency Raman spectra are scaled for the clarity.

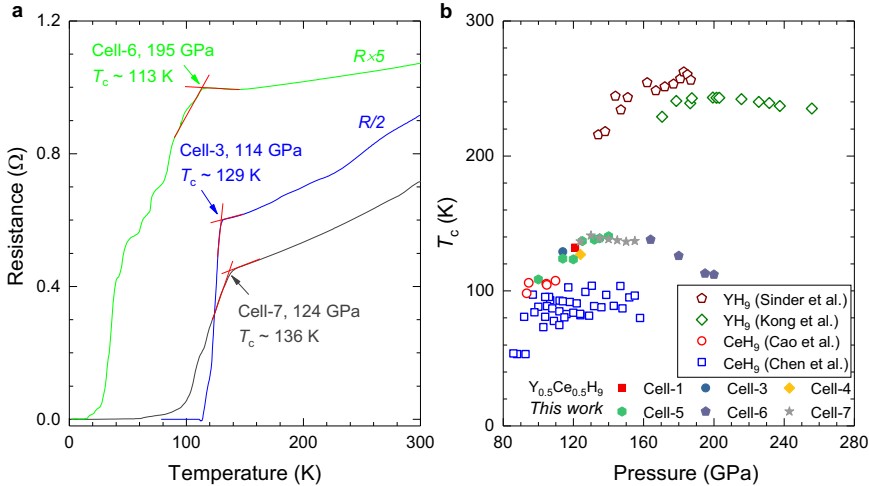

**Fig. 2 | Temperature-dependent resistance measurements on the synthesized $Y_{0.5}Ce_{0.5}$ hydrides in the representative cells. a** Superconducting transitions at various pressures were indicated by the arrows. **b** Pressure dependence of $T_c$ for $Y_{0.5}Ce_{0.5}H_9$ with the comparison of $YH_9$ and $CeH_9$ with the same crystal structure in the literature[10,12,17,18]. The sold symbols with different colours represent the results from six runs in the current work. The open symbols denote the data points for $YH_9$ from the works[10,12] and for $CeH_9$ from the works[17,18], respectively.

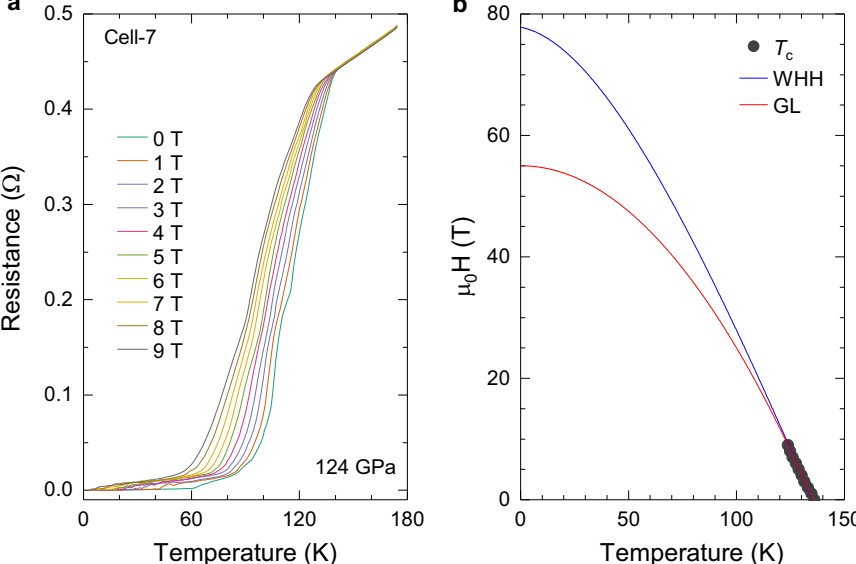

**Fig. 3 | Electrical transport measurements of the synthesized $Y_{0.5}Ce_{0.5}$ hydrides at external magnetic fields in Cell-7. a** Temperature dependence of the resistance at applied magnetic fields from 0 to 9 T at pressure of around 124 GPa. **b** Upper critical field $H_{c2}$ as a function of temperature, fitted with the GL and WHH model, respectively.

To further obtain the other superconducting properties, we performed the electrical transport measurements at external magnetic fields for sample 7 (Cell-7) at pressure of 124 GPa (Fig. 3). Supplementary Fig. 8a shows the Raman spectra of diamond used for the pressure calibration[28]. As shown in Fig. 3a, with the application of magnetic field up to 9 T, the resistance gradually shifts to lower temperatures with the widening of the superconducting transition. The suppression of superconductivity by the applied fields can be clearly observed. These phenomena are the character of superconductivity. The temperature dependence of the upper critical field $\mu_0H_{c2}$ can be obtained from such measurements (Fig. 3b). The upper critical field is usually described by the Ginzburg-Landau (GL) equation[29] and the Werthamer-Helfand-Hohenberg (WHH) equation[30].

The upper critical field $\mu_0H_{c2}$(T) at temperature of 0 K is extrapolated to be 56 and 78 Tesla from the GL and WHH model fitting, respectively. The achievement of the zero-resistance state

together with the downward shift of $T_c$ and gradual widening of the superconducting transition with applied magnetic fields demonstrate the superconductivity in synthesized samples of $Y_{0.5}Ce_{0.5}$ hydrides. The coherence length $\xi$ can be roughly calculated via the equation of $\mu_0H_{c2}(0) = \phi_0/2\pi\xi^2$, where $\phi_0$ is the magnetic flux quantum. By using the obtained $\mu_0H_{c2}(0)$ of 56-78 Tesla, one can roughly estimate $\xi$ to be 20-25 Å. The high upper critical fields and short coherence lengths suggest the type-II character of the synthesized superconductors.

### Structure characterization

Synchrotron x-ray diffraction (XRD) measurements were carried out on sample 1 (Cell-1) and sample 5 (Cell-5) to determine their structural properties. For sample 1, the representative XRD patterns are given in Fig. 4a for Point-1 and Supplementary Fig. 9 for Point-2, respectively, at pressure of 128 GPa determined from the Raman spectrum of the used

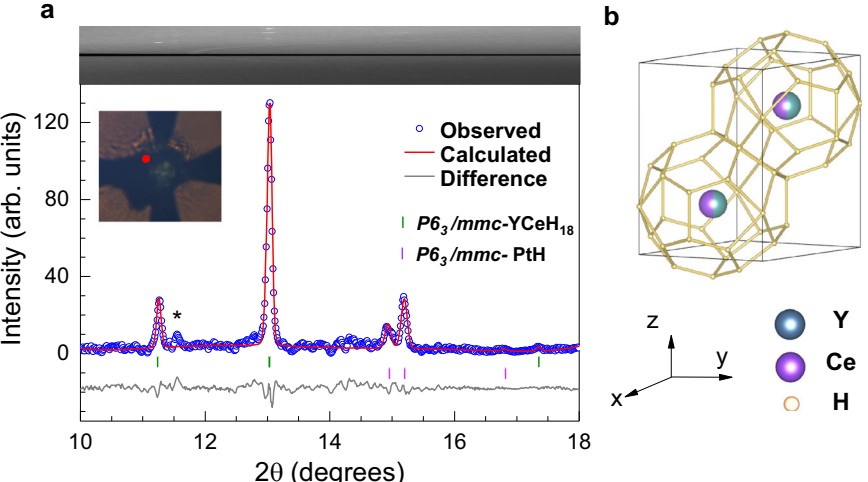

**Fig. 4 | Synchrotron x-ray diffraction patterns of sample 1 at pressure of 128 GPa (Point-1). a** The experimental data along with the model fitting for the phases of $P6_3/mmc$-$Y_{0.5}Ce_{0.5}H_9$ and $P6_3/mmc$-PtH. The experimental data points, calculated values, and Bragg peak positions are shown by the small open circles, thin curves, and vertical sticks, respectively. Top panel shows the cake view of the raw XRD patterns. The red point in the inset photograph displays the corresponding sample position of the collected patterns. The marked peak with asterisk represents the unidentified reflection. **b** Schematic diagram of the crystal structure of $P6_3/mmc$-$Y_{0.5}Ce_{0.5}H_9$.

diamond anvil (Supplementary Fig. 10). Comparing the structural features of the similar system in the literature[9,10,12,17,25], we have the phases with the structures in the space groups of $P6_3/mmc$, $I4/mmm$, and $C2/m$ for the syntheized products. As shown in Fig. 4a, the observed peaks collected at the red dot in the inset can be indexed by the $P6_3/mmc$ phase with the help of the Le Bail method[31]. The fitted lattice parameters are $a = 3.653(5)$ Å, $c = 5.476(5)$ Å, and $V = 63.33(4)$ Å$^3$ (Supplementary Table 2). The weak diffraction peaks at larger angles are from PtH-II ($P6_3/mmc$) as reported previously[32], which is common in the chemical reaction of Pt leads with hydrogen[32,33]. Other representative XRD patterns (Supplementary Fig. 9) correspond to three $Y_{0.5}Ce_{0.5}$ hydride phases with the space groups of $P6_3/mmc$, $I4/mmm$, and $C2/m$, and the Pt hydride phase of $P6_3/mmc$.

To draw more structural information, Cell-5 with large diffraction angles was prepared for synchrotron XRD measurements. Typical results are given in Supplementary Figs. 11–13. By indexing the observed peaks of Sample 5 at 140 GPa, we can identify the appearance of three phases with the $P6_3/mmc$, $I4/mmm$, and $C2/m$ structure for $Y_{0.5}Ce_{0.5}$ alloy hydrides and the $P6_3/mmc$ PtH-II. The XRD patterns of pure $Y_{0.5}Ce_{0.5}$ phase and pure PtH-II phase in Cell-5 at 140 GPa were also collected (Supplementary Fig. 13). Although the accurate occupancy of the hydrogen atoms can not be determined experimentally owing to the weak x-ray scattering cross sections, the hydrogen concentration ($x$) can be obtained through the cell unit volume ($V_{lattice}$) by using the formula $x = (V_{lattice} - V_Y - V_{Ce})/V_H/2$, where $V_Y$, $V_{Ce}$, $V_H$ are the volumes of Y, Ce, and H atoms, respectively. The $V_Y$, $V_{Ce}$, and $V_H$ can be estimated from their elemental phases (Supplementary Fig. 14)[34–36] because of the absence of the atomic volume at high pressures. As a result, the hydrogen content of the synthesized $P6_3/mmc$ phase of $Y_{0.5}Ce_{0.5}$ alloy hydrides is in the range of 8.3–8.8 (Supplementary Table 2). The calculated hydrogen concentration has a small deviation with the ideal value of 9, because of the direct use of the elemental phases for the evaluation of volumes.

## Superconducting phase from theoretical calculations

To identify the obtained superconducting phase, we calculated the electronic, phononic, and superconducting properties of $P6_3/mmc$-$Y_{0.5}Ce_{0.5}H_9$ in terms of the density functional theory (Fig. 5). The minimum pressure that can theoretically stabilize the structure $P6_3/mmc$-$Y_{0.5}Ce_{0.5}H_9$ is 180 GPa. As shown in Fig. 5a, the $P6_3/mmc$ structure

is metallic and shows a large density of states (DOS) near the Fermi level $E_F$. The large DOS is in favor of high-$T_c$ superconductivity. The contributions of the electronic states of H near $E_F$ usually plays an important role for the hydride superconductor. From the partial DOS shown in Fig. 5a, we can see that the large DOS near $E_F$ is dominant from the contribution of the H-$s$, Ce-$f$, and Y-$d$ orbitals. In detail, the contributions of the electronic states of H-$s$, Ce-$f$ near $E_F$ are both 0.37 states/eV. The large contribution of Ce-$f$ indicates that the delocalized nature of Ce-$f$ electrons is responsible for the enhanced chemical precompression in $Y_{0.5}Ce_{0.5}$ hydrides.

The calculated phonon dispersion, phonon density of states (PHDOS), Eliashberg function $\alpha^2F(\omega)$, and electron-phonon coupling (EPC) constant $\lambda(\omega)$ of $P6_3/mmc$-$Y_{0.5}Ce_{0.5}H_9$ are given in Fig. 5b. The optical phonon modes at high frequencies (above ~390 cm$^{-1}$) are derived from the H atoms. The acoustic phonon modes with frequencies lower than ~180 cm$^{-1}$ and the low-frequency optical modes between 145 and 320 cm$^{-1}$ are mainly attributed to the contributions of Y and Ce atoms. From the phonon frequency dependence of $\alpha^2F(\omega)$ and integrated EPC constant $\lambda(\omega)$, we find that the contributions to $\alpha^2F(\omega)$ and $\lambda(\omega)$ arise from all three phonon modes including the Y-derived acoustic, the Ce-derived acoustic, and the H-derived optical modes. In addition, the wide high-frequency $\alpha^2F(\omega)$ is significantly higher than those on the narrow low-frequency side. Obviously, the high-energy H-derived vibrations dominate the total $\lambda(\omega)$ value. The calculations give a high EPC constant $\lambda$ of 2.23 at 180 GPa. By numerically solving the Allen-Dynes-modified McMillan formula with the Coulomb pseudopotential parameter $\mu^*$ of 0.1–0.15[37,38], we obtained $T_c$ of 104–119 K (Supplementary Table 3), in fair agreement with our experiments. The absence of the imaginary parts of the phonon spectrum (Fig. 5b) along with the comparable $T_c$ values with experiments supports the obtained superconducting phase of $P6_3/mmc$-$Y_{0.5}Ce_{0.5}H_9$.

Compared the synthetic conditions, stable pressures, and $T_c$ values of $P6_3/mmc$-$Y_{0.5}Ce_{0.5}H_9$ with $YH_9$ and $CeH_9$ (Fig. 2b and Supplementary Table 4), we found that the synthetic pressure and minimum stable pressure of $Y_{0.5}Ce_{0.5}H_9$ are much lower those of $YH_9$[10,12], and the $T_c$ values are overall higher than those of $CeH_9$[17,18] at around the same pressure level. $Y_{0.5}Ce_{0.5}H_9$ possesses the maximum of $T_c$ of 141 K at the optimal pressure of about 130 GPa. This $T_c$ value is apparently higher than all the reported values for $CeH_9$[17,18]. The optimal pressure is

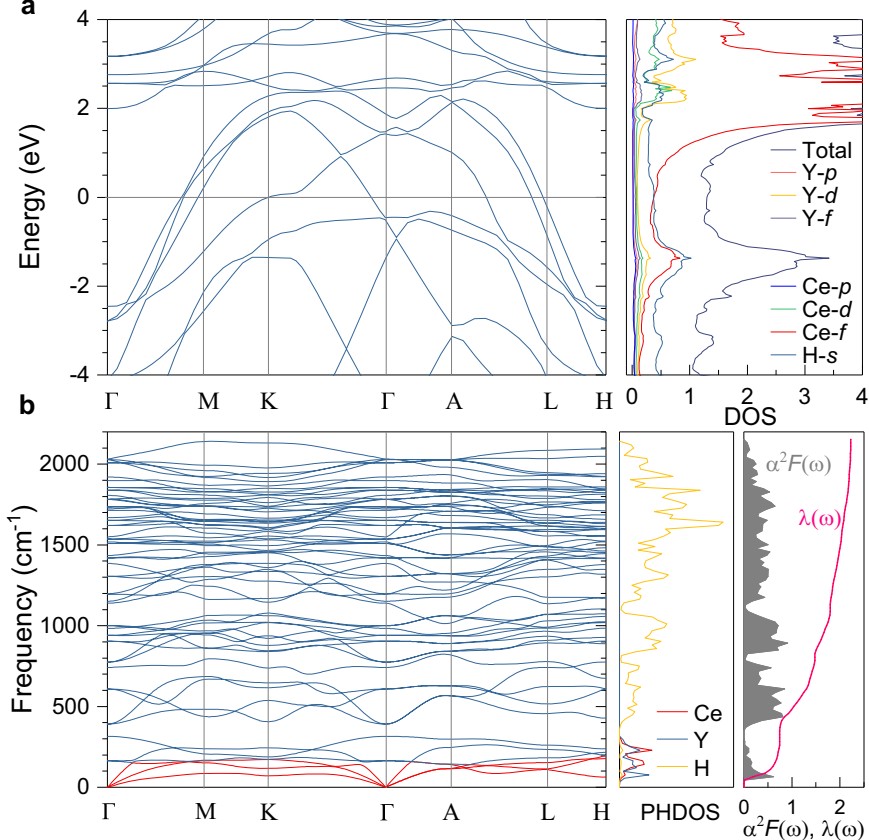

**Fig. 5 | The electronic and phononic properties of $P6_3/mmc$ $Y_{0.5}Ce_{0.5}H_9$ at 180 GPa. a** Band structure, total and partial electron density of states (DOS). The horizontal line shows the location of the Fermi level. **b** Calculated phonon dispersion, phonon density of states (PHDOS) projected onto the Y, Ce, and H atoms, Eliashberg function $\alpha^2F(\omega)$, and integrated electron-phonon coupling (EPC) constant $\lambda(\omega)$.

much lower than those of $YH_9$[10,12] from two sets of data points in independent runs. $Y_{0.5}Ce_{0.5}H_9$ thus serves an excellent example for the successful realization of high-$T_c$-superconductivity in ternary alloy hydrides. The comparison of our results for $Y_{0.5}Ce_{0.5}H_9$ with those for $(La,Ce)H_9$ and $La_{0.5}Ce_{0.5}H_{10}$ taken from the recent works[25,33] was given in Supplementary Fig. 15 as well. It is clear that the hydrides based on the alloy including Ce and other rare-earth element can substantially reduce the optimal pressure while keeping higher $T_c$ compared to Ce-based hydrides.

In summary, we have successfully synthesized $Y_{0.5}Ce_{0.5}$ hydrides by laser heating at high pressures in seven runs (Cell-1 to Cell-7). The synthesized $Y_{0.5}Ce_{0.5}H_9$ with high $T_c$ of 97–141 K in the pressure range of 98–200 GPa was comprehensively studied by the measurements of the electrical transport, x-ray diffraction, and Raman scattering, and theoretical calculations. Compared with the binary cerium hydrides $(CeH_9)$[17,18], this half-replaced ternary alloy superhydride $(Y_{0.5}Ce_{0.5}H_9)$ possesses significantly high $T_c$ values. Our findings point out the promising direction for exploring high $T_c$ superconductivity under pressure in alloy hydrides. In addition, the detailed synthesis process of the yttrium-cerium hydrides hopes to help subsequent investigations for other hydrides.

## Methods
### Sample synthesis
The $Y_{0.5}Ce_{0.5}$ alloy ingots were prepared by melting pure Y (99.9%, Alfa Aesar) and Ce (99.8%, Alfa Aesar) in an arc-melting machine under a Ti-gettered high purity argon atmosphere. To ensure uniformity of alloy ingot, we flipped each ingot and melted it at least six times. $Y_{0.5}Ce_{0.5}$ hydrides were synthesized by using a series of symmetric diamond-anvil cells (DACs), which have diamonds bevelled at 8–8.5° to a

diameter of 300 µm with a culet size of 60–100 µm. In detail, we employed the ammonia borane ($NH_3BH_3$ or AB) as the hydrogen source, which has been successfully used in several recent studies[3,17]. When being completely dehydrogenated, one mole of AB can yield three moles of $H_2$ along with the insulating c-BN by the decomposing reaction $NH_3BH_3 \rightarrow 3H_2 + $ c-BN[39,40]. The $Y_{0.5}Ce_{0.5}$ pieces were cut into 15–30 µm in diameter and 1–2 µm thick. The sheared pieces and AB were then layered into the sample chamber. To isolate the surrounding atmosphere, we handled all the samples in a glove box with the residual contents of $O_2$ and $H_2O$ of <0.1 p.p.m. One-side laser-heating experiments were performed using a yttrium-aluminum-garnet (YAG) laser, after the pressure was increased to the desirable value. The heating temperature is between 1500–2000 K (Supplementary Table 1) and the exposure time is 10–20 min. In our experiments, we used the laser power of 5 mW to focus on the sample with the laser spot around 1.5 micrometer.

### Raman scattering measurements
Raman scattering measurements were used to characterize the synthesizing processes of $Y_{0.5}Ce_{0.5}$ hydrides. The measurements were performed with the exciting laser (488 nm) and the beam size of the laser of about 3 µm. The scattered light was focused on a 300 and 1800 g/mm grating and recorded with a 1300 pixel charge-coupled device designed by Princeton Instrument. In all the experiments, the pressure was determined using the Raman shift of the stressed edge of the diamond peak[28].

### Electrical transport measurements
A typical diamond anvil cell made from Cu-Be alloy was used for the electronic transport measurements at external magnetic fields from

0 to 9 T. Four Pt wires were adhered between the $Y_{0.5}Ce_{0.5}$ alloy and AB in the sample hole with silver epoxy (Supplementary Fig. 1) by the Van Der Pauw method[41] to measure the in-plane resistance at high pressures. This technique was successfully used to measure the resistivity and Hall resistivity in the study of superconductivity for materials in hydrostatic or quasi-hydrostatic pressure environments[18,33,42–46]. In the present resistance measurements, the rhenium gaskets insulated by the mixture of cubic boron nitride (*c*-BN) and epoxy were used to contain the sample at megabar pressures while isolating the platinum (Pt) electrical leads. The whole progress was performed in the glove box. Since hydrogen produced from the decomposed AB serves as the pressure transmitting medium, the technique developed here and previously[18,33] for hydrides ensures the resistance measurements in the excellent hydrostatic pressure environments at the studied pressures.

## X-ray diffraction measurements

The structural characterization was carried out at Shanghai Synchrotron Radiation Facility with a wavelength 0.6199 Å and the size of the x-ray focus beam is less than $2\,\mu m$. Considering the possible inhomogeneity of the synthesized size, we collected the synchrotron XRD patterns with $5\,\mu m$ step across the culet. The obtained two-dimensional XRD patterns are converted to one dimensional diffraction data with the help of Dioptas[47]. The data was analyzed by using the software of Jana based on the Le Bail method[31,48]. The fitted lattice parameters are summarized in the Supplementary Information.

## Theoretical calculations

For the theoretical calculations, the structure was fully relaxed before calculating the electronic band structures. The electronic band structures were calculated by using Vienna ab initio simulation package (VASP)[49,50] with the density functional theory (DFT)[51,52]. The generalized gradient approximation (GGA) of Perdew-Burke-Ernzerhof (PBE) functional was used for the exchange-correlation functional[53]. The cut-off energy of the plane-wave basis was set to 400 eV. The phonon spectrum, electron-phonon coupling (EPC) parameters, and $T_c$ were carried out using the QUANTUM ESPRESSO (QE) package[54] by employing the plane wave pseudopotential method and PBE exchange-correlation functional[53]. The selected *k*-point mesh in this part is $12 \times 12 \times 12$, and the *q*-point mesh is $3 \times 3 \times 3$.

# Data availability

The data that support the findings of this study have been included in the plotted figures and listed tables in the main text and Supplementary Information. If any additional information or any extra data will be required in order to reproduce the results reported in this work, please contact the corresponding author.

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

## Acknowledgements

The work was supported by the Basic Research Program of Shenzhen (Grant No. JCYJ20200109112810241), the Shenzhen Science and Technology Program (Grant Nos. KQTD20200820113045081 and RCBS20221008093127072), the National Post-doctoral Program for Innovative Talents (Grant No. BX2021091), and the National Key R&D Program of China (Grant No. 2018YFA0305900).

## Author contributions

X.J.C. designed and supervised the whole study. D.P. synthesized the $Y_{0.5}Ce_{0.5}$ alloy. L.C.C., Z.Y.C., and G.H. synthesized Y-Ce-H samples. L.C.C. performed high-pressure resistance and Raman scattering measurements under the guidance of X.J.C. L.C.C. and L.L.Z. collected the XRD data. T.L., G.H.Z., and H.Q.L. carried out the theoretical calculations. P.D.S. and F.A.G. help to heat the Cell-1. L.C.C. and X.J.C. analyzed the data and wrote the paper with the inputs from other authors. All authors discussed the results. The manuscript reflects the contributions of all authors.

## Competing interests

The authors declare no competing interests.
