## [Peer Review File · Nature Communications]

Synthesis and superconductivity in yttrium-cerium hydrides at high pressuresREVIEWER COMMENTS

Reviewer #1 (Remarks to the Author):

This is the review report for the manuscript 380516_0 entitled Synthesis and superconductivity in yttrium-cerium hydrides at moderate pressures submitted by Liu-Cheng Chen et al.

This manuscript reports the synthesis of the ternary hydrides $Y_{0.5}Ce_{0.5}H_9$ under pressure and the observation of the superconductivity with superconducting transition temperatures (T_c s) ranging from 97 to 140 K by electrical resistance measurements. The observed T_c s are higher than the previously reported CeH_9 and lower than YH_9 . The significant point is that the maximum T_c near 140 K appears at lower pressures than YH_9 by around 60 GPa. The X-ray diffraction measurements observed three structural phases (P63/mmc, I4/mmm, C2/m). Raman scattering measurements and the theoretical calculation suggest that the P63/mmc phase is the high- T_c superconducting phase.

Among the intense competition to synthesize high- T_c hydrides, like the previously reported superconducting $La_{0.5}Ce_{0.5}H_9$ with $T_c = 178$ K, the present study can be one of the milestones in the field. This reviewer highly evaluates the success in synthesizing ternary hydride $Y_{0.5}Ce_{0.5}H_9$ with a high- T_c exceeding 140 K. Another significant point of this study is achieving the high- T_c at a lower pressure than YH_9 , suggesting the lowering of stability pressure regime by substituting Y with Ce.

While highly evaluating the experimental achievements, this reviewer has a concern about the mismatch between the study's purpose and design. Also, this reviewer thinks the manuscript needs more discussion about the crystal structure of the high- T_c phase.

This reviewer would like to invite the authors to address the questions and criticisms listed below.

Questions and criticisms

1. Mismatch between the study's purpose and design of the study.

The main conclusion indicated by the authors is the synthesis of $Y_{0.5}Ce_{0.5}H_9$ at moderate pressure, claiming the ternary compound has a minimum stable pressure lower than YH_9 and other YH_x . However, the present study has not investigated the stable pressure region and required synthesis conditions (P and T). The authors performed four runs of the HP/HT synthesis of $Y_{0.5}Ce_{0.5}H_9$. The pressure was almost fixed in each run. The temperature during the laser heating is not described. When one wants to find the stability region and the minimum pressure required for the synthesis, it is necessary to try the synthesis at various temperature-pressure conditions. Besides, the authors need to release/increase the applied pressure and find the decomposition pressure.

Has any experimental study investigated the stable pressure region of YH_x ($x > 4$) and CeH_x superconductors comprehensively covering the wide temperature and pressure region? The pressures indicated in Fig. 4c are values where the highest T_c s has been reported to date, neither the decomposition pressures nor the pressure where T_c s reach maximum. Also, the pressure dependence of the T_c s for those superhydrides has not been revealed yet.

Therefore, it is not scientifically rigorous to claim that $Y_{0.5}Ce_{0.5}H_9$ can be synthesized at 'moderate' pressure and stable at a lower pressure than YH_x without comparing the stability region and synthesis conditions.

2. Discussion about the high- T_c phase.

After laser heating, the Raman scattering measurements observed a new peak near 125 cm^{-1} . Based on their theoretical calculation, the authors attribute the peak to the hexagonal P63/mmc phase of $Y_{0.5}Ce_{0.5}H_9$. However, observing just one phonon mode does not ensure that it is from the P63/mmc phase.

In the sample chamber of DAC, there are H_2 , c-BN, NH_3BH_3 , Pt, and carbon (diamond anvils). NH_3BH_3 decomposes into various phases, including c-BN and di-borane, depending on the applied

temperature and pressure. Pt reacts with H₂ forming PtH in P63/mmc structure and a superconductor. The authors need to discuss if those substances and their compounds caused by heating have a chance to show Raman active phonon modes at 125 cm⁻¹. The position where the excitation laser was focused would be important information.

Suppose it is impossible to conclude that Y_{0.5}Ce_{0.5}H₉ is the only possible origin of the peak. In that case, this reviewer recommends rephrasing the sentences, such as 'Our theoretical calculation suggests that the peak marked asterisk is the optical phonon mode of the synthesized sample.' Since the present study can suggest the crystal structures of Y_{0.5}Ce_{0.5}H₉, it would be possible to compare them with the previously reported YH₉, CeH₉, LaBH₈, and La_{0.5}Ce_{0.5}H₉. The comparison should lead to the discussion of which structure is preferable for high-T_c.

In the Results and Discussions for sample #4, the authors consider that the inhomogeneous sample does not allow the resistance to drop to zero. The authors should elaborate on what 'inhomogeneous' mean. Is the sample the mixed phase of P63/mmc, I4/mmm, and C2/m? Does it mean that the hydrogen content varies? Does the 'inhomogeneous' differ from the 'incomplete reaction' mentioned for sample #1?

3. Notation of the synthesized sample Y_{0.5}Ce_{0.5}H₉

Page 6. Since V vs. P data (Fig. 4b) does not give the precise value of hydrogen content, it would be appropriate to write 'the hydrogen content of the P63/mmc phase is close to 9.' The same comment is applied to I4/mmm phase. It is well known that hydrides consist of a large number of defects.

4. Page 4, the experiment of sample #1. 'However, we have not reached zero resistance due to the incomplete reaction.' Is there any experimental evidence to support the 'incomplete reaction'?

5. Page 6. 'Similar results have been discussed in LaH_x and LaH₆₋₇ with the same crystal structure 5,23.' What this sentence tries to convey is not clear.

6. Page 6. The lattice constants of Y_{0.5}Ce_{0.5}H₉ are indicated. The angle of the C2/m phase is missing.

7. Organization of the manuscript

This referee recommends reorganizing the manuscript for readability and avoiding confusion. More specifically, it would be better to switch the position of samples #4 and #1. When this reviewer saw Fig. 1, particularly the Raman scattering spectrum, this reviewer wondered why the authors could identify the marked (*) asterisk from the sample. Also, similar questions arise, such as 'Is the peak at 125 cm⁻¹ a peak? Isn't it a noise?' The results of #4 are cleaner and more convincing that the new peak appears after heating.

8. Experimental information

The present manuscript tries to provide every single detail of experimental conditions. This reviewer personally truly likes such kind of sincere attitude. However, the manuscript lacks the temperature during laser heating. If the authors did not measure the temperature, it should be mentioned. It also would be a good idea to write the machine model name of the Raman scattering measurement system. It is a good reference for readers.

9. Raman scattering peak analysis

The multiple peak fitting shown in Fig. 1b lower panel looks strange. S(1) peak is symmetric. On the other hand, S(0) is asymmetric. Besides, S(0) seems to include a background. E_{2g} is not fitted. How did the authors assume the peak shapes in their fitting? How was a background handled? Did the authors conduct the multiple peak fitting using all four peaks?

Minor points

Page 6. 'Thus, the successfully synthesized Y_{0.5}Ce_{0.5}H₉ hope to advance the search of alloy hydrides with high-T_c at relatively moderate pressure.' The underlined part should be like 'suggests the hope.'

Page 8. 'The $Y_{0.5}Ce_{0.5}$ pieces were cut into 15-' were -> were

Fig. 1a, 2a, 3a, S1. The pictures placed side-by-side should be aligned in the same orientation to make it easier to see the samples' shape changes.

The title of Fig. S1 must be wrong. Run 2 did not observe the superconductivity due to the technical problem.

Reviewer #2 (Remarks to the Author):

This paper reports in situ measurements of Y:Ce alloys loaded with ammonia borane (AB) inside a DAC. These samples were each then taken to pressures in excess of 110 GPa and then laser heated. Visual expansion of all samples was reported, and in situ transport measurements were conducted on three of the samples. Of these, sample #S1 retained a finite resistance down to 0K, while samples #S3 and #S4 reached zero resistance. These observations are interpreted as the formation of Y:Ce hydride, which becomes superconducting upon cooling (the residual resistance of #S1 is explained to indicate partial synthesis of the super-conducting phase).

The authors observe a fairly clear optical change in their sample, which appears to grow larger after the laser-induced reaction in at least two of the sample for which images are shown (#S4 shows little change). Raman data are shown on the reacted phase, but these show little beyond the presence of H₂, evolved from the AB. A very weak, low wavenumber feature seen around 120 cm⁻¹ is attributed to a phonon mode of Y:Ce super hydride. However, this yields little information on the nature or composition of the hydride: there is certainly no clear difference between “incomplete” reacted sample #S1 and the fully reacted #S3 and #S4.

The increased size of the sample is interpreted as hydride formation, however, this is very qualitative. The same effect could also result from gasket thinning causing the sample to thin, thus increasing its area without increasing its volume. Accordingly, since the Raman is also ambiguous, the experimental part of this work really hangs on the synchrotron x-ray diffraction data presented. Moreover, due to the lack of x-ray radiation's lack of sensitivity to hydrogen locations, and the high degree of texture in the sample (evident from the 2d data shown at the top of Fig 4a), the only information that might reliably be extracted is the volume of the unit cell. The authors are aware of this limitation and have used the Le Bail analysis approach, in which the Bragg peak intensities are unconstrained by any structural model.

Unfortunately, I have some serious concerns with the analysis of the diffraction data. Firstly, it's a shame that diffraction data are only shown for sample #S1, which is only partly reacted and does not show a full SC transition. The partial reaction presumably means we should expect unreacted Y:Ce metal to be potentially present. In addition, although they it will scatter rather weakly, there may be some c-BN present as a bi-product of the AB de-hydrogenation. However, neither possibility seems to have been considered in the analysis presented.

The sample is analysed as being comprised of a mixture of three hydride phases with space groups: P6₃/mmc (hcp), I4/mmm (tetragonal) and c2/m (monoclinic). This already provides a lot of flexibility to fit the observed diffraction pattern but clearly fails to describe the observations between 15-16 degrees: where only a single peak (from the monoclinic phase) is predicted, two peaks are clearly observed (Fig A blue arrows).

In addition, the largest diffraction feature, seen at 14 degrees, to my eye has more structure (at least 3 peaks) than the two available peaks in the model (one tetragonal one monoclinic) allow. This structure is hidden by the chosen x-axis scaling, but is revealed in the difference curve. I have tried to highlight by changing the aspect ratio. The inflections are quite clearly visible (Fig B blue arrows) and I think would be obvious if a greater zoom-range were used in this region. Lastly, I had a similar observation around 17 degrees, where there is a clear, unindexed shoulder (C blue arrow) on the left-hand side of the peak.

The conclusion I draw from these observations is that there are issues with the indexing (i.e. allocation of observed intensity to model peaks) and most likely additional phases in the pattern that are not correctly treated.

A second technical issue is observed with the peak width for the tetragonal phase. The Le Bail refinement is clearly strongly steered towards fitting the isolated peak just above 12 degrees. This peak is well fitted, but the other peaks due to the same phase will be constrained to have similar width. This clearly misfits the data at 18 degrees (Fig C red arrow), so I am left wondering about the allocation of intensity in the main peak at 14 degrees.

These deficiencies in the fit to the observations, to me, leave significant question marks over the extracted unit cell volumes and, since these are the main experimental basis of the manuscript, weaken it significantly.

A final criticism I had is with the inclusion of the crystal structures as insets to Figure 4a. These imply that a definitive structure is known when, in fact, the observations only yield heavy atom positions and an inferred (and, likely inaccurate, on the basis of above comments) estimate of hydrogenation from volumetric measurements. So, how are the hydrogen sites known? Whether these are taken from calculation, from citation, inferred, or just guessed is not clear as the caption merely states "inset shows the crystal structure". There is also inaccuracy in the colour coding of the Ce and Y atoms. These are shown to separately occupy equivalent atomic sites in the crystal structure, making them inequivalent. Thus, the figures are not consistent with the space groups given. In reality, these species are substitutionally disordered over the *same* crystallographic site in a way that the crystal structure images misrepresent.

The apparently clear observation of SC under these pressure-temperature conditions is certainly significant and of broad interest. However, exactly what was synthesised remains ambiguous on the basis of the data and analysis presented. Unfortunately, this means I can't recommend publication of this manuscript in its present form.

Reviewer #3 (Remarks to the Author):

The manuscript presents the synthesis, phase analysis, and electrical transport data that suggest the appearance of superconductivity in yttrium-cerium hydrides at 110-120 GPa pressures. The experimental part is complemented by band structure calculations.

The topic of the manuscript is interesting and timely.

I do realize that any measurements in the megabar pressure range are extremely difficult and require a lot of effort. This said, the presented transport data do not convince the reader that there is bulk superconductivity in the samples (in principle, one can have zero resistance with filamentary superconductivity associated with some minority phase). The author did not report any attempts at further measurements that would make this claim stronger: resistance in a magnetic field, I-V curves, ac susceptibility, dc susceptibility, etc...

Having in mind these limitations, I cannot recommend publication of this work in Nature Communications.

Summary of change on manuscript # NCOMMS-22-29561

We have substantially revised the manuscript in accordance with all the comments and suggestions from three reviewers. The detailed changes are summarized as follows:

1. The manuscript has been organized based on the results of seven independent samples (Cell-1 to Cell-7).
2. Three tables (synthesis conditions, measurements, XRD parameters, and comparisons between YH_x , CeH_9 and $\text{Y}_{0.5}\text{Ce}_{0.5}\text{H}_9$) have been provided in the supplementary materials.
3. To obtain the pressure-dependent T_c , we performed three more runs of the measurements for other three cells (Fig. 2-3 and supplementary Fig. S6).
4. More details about the structures and phases have been given with the XRD cell (Cell-6) in revised manuscript (Fig. 4 and supplementary Figs. S9-12).
5. To confirm the superconducting state, we provided the data from the electrical transport measurements under external magnetic field (Fig. 3).
6. All the minor errors pointed by the reviewers have been corrected, such as the Raman spectrum analysis, unclear expression, uncorrected sentences and words.
7. The author named Li-Li Zhang has been added due to his help of the XRD measurements.

Response to the first reviewer's comments on NCOMMS-22-29561

Comment 1:

“Mismatch between the study's purpose and design of the study. The main conclusion indicated by the authors is the synthesis of $Y_{0.5}Ce_{0.5}H_9$ at moderate pressure, claiming the ternary compound has a minimum stable pressure lower than YH_9 and other YH_x . However, the present study has not investigated the stable pressure region and required synthesis conditions (P and T). The authors performed four runs of the HP/HT synthesis of $Y_{0.5}Ce_{0.5}H_9$. The pressure was almost fixed in each run. The temperature during the laser heating is not described. When one wants to find the stability region and the minimum pressure required for the synthesis, it is necessary to try the synthesis at various temperature-pressure conditions. Besides, the authors need to release/increase the applied pressure and find the decomposition pressure.”

Reply: Many thanks for the reviewer's comments. We studied the yttrium-cerium hydrides in a wide pressure range from 98 GPa to 166 GPa in the revised manuscript. The synthesis conditions (P and T) for the yttrium-cerium hydrides are summarized in the supplementary Table S1.

“Has any experimental study investigated the stable pressure region of YH_x ($x > 4$) and CeH_x superconductors comprehensively covering the wide temperature and pressure region? The pressures indicated in Fig. 4c are values where the highest T_c s has been reported to date, neither the decomposition pressures nor the pressure where T_c s reach maximum. Also, the pressure dependence of the T_c s for those superhydrides has not been revealed yet. Therefore, it is not scientifically rigorous to claim that $Y_{0.5}Ce_{0.5}H_9$ can be synthesized at 'moderate' pressure and stable at a lower pressure than YH_x without comparing the stability region and synthesis conditions.”

Reply: Many thanks for the reviewer's comments. The comparisons of synthetic pressure, stable pressure and T_c of YH_x ($x > 4$), CeH_9 , and $Y_{0.5}Ce_{0.5}H_9$ are summarized in the supplementary Table S4. Due to the different stability region and synthesis conditions between YH_x and $Y_{0.5}Ce_{0.5}H_9$, we thus provided the detailed comparison for them.

Comment 2:

Discussion about the high- T_c phase. After laser heating, the Raman scattering measurements observed a new peak near 125 cm^{-1} . Based on their theoretical calculation, the authors attribute the peak to the hexagonal P63/mmc phase of

$Y_{0.5}Ce_{0.5}H_9$. However, observing just one phonon mode does not ensure that it is from the $P6_3/mmc$ phase. In the sample chamber of DAC, there are H_2 , c-BN, NH_3BH_3 , Pt, and carbon (diamond anvils). NH_3BH_3 decomposes into various phases, including c-BN and di-borane, depending on the applied temperature and pressure. Pt reacts with H_2 forming PtH in $P6_3/mmc$ structure and a superconductor. The authors need to discuss if those substances and their compounds caused by heating have a chance to show Raman active phonon modes at 125 cm^{-1} . The position where the excitation laser was focused would be important information. Suppose it is impossible to conclude that $Y_{0.5}Ce_{0.5}H_9$ is the only possible origin of the peak. In that case, this reviewer recommends rephrasing the sentences, such as 'Our theoretical calculation suggests that the peak marked asterisk is the optical phonon mode of the synthesized sample.'

Reply: Many thanks for the reviewer's comments and suggestions. As shown in the Fig. 1a of the revised manuscript, the exciting laser direction of the Raman scattering measurement is from the alloy side, thus it is unlikely that the collected Raman spectra are from c-BN, NH_3BH_3 , Pt, or PtH. In addition, the Raman peak marked with asterisk should not be from diamond or H_2 , due to the absence of any Raman active modes for them in this region at the studied pressure. Therefore, we attribute the marked peaks to the optical phonon mode of the synthesized $Y_{0.5}Ce_{0.5}$ hydrides in the revised manuscript.

Since the present study can suggest the crystal structures of $Y_{0.5}Ce_{0.5}H_9$, it would be possible to compare them with the previously reported YH_9 , CeH_9 , $LaBH_8$, and $La_{0.5}Ce_{0.5}H_9$. The comparison should lead to the discussion of which structure is preferable for high- T_c .

Reply: Many thanks for the reviewer's comments and suggestions. Theoretical calculation indicates that $Fm\bar{3}m-LaBH_8$ is dynamically stable at low pressures with T_c up to 126 K at 50 GPa. However, it was not reported in the experiments. Depending on the experimental report, YH_9 , CeH_9 , and $La_{0.5}Ce_{0.5}H_9$ share the same structural phase ($P6_3/mmc$), which are consisted with $Y_{0.5}Ce_{0.5}H_9$ studied in this work.

In the Results and Discussions for sample #4, the authors consider that the inhomogeneous sample does not allow the resistance to drop to zero. The authors should elaborate on what 'inhomogeneous' mean. Is the sample the mixed phase of $P6_3/mmc$, $I4/mmm$, and $C2/m$? Does it mean that the hydrogen content varies? Does the 'inhomogeneous' differ from the 'incomplete reaction' mentioned for sample #1?

Reply: Many thanks for the reviewer's comments. We have changed "the possible

Inhomogeneous superconducting phases” to “the superconducting phases with different hydrogen amount” in revised manuscript.

Comment 3: Notation of the synthesized sample $Y_{0.5}Ce_{0.5}H_9$. Page 6. Since V vs. P data (Fig. 4b) does not give the precise value of hydrogen content, it would be appropriate to write ‘the hydrogen content of the P63/mmc phase is close to 9.’ The same comment is applied to I4/mmm phase. It is well known that hydrides consist of a large number of defects.

Reply: *Many thanks for the reviewer’s comments. In the revised manuscript, we experimentally obtain the hydrogen concentration (x) through the cell unit volume. The hydrogen content of the synthesized $Y_{0.5}Ce_{0.5}$ alloy hydrid was determined as 8.3-8.8.*

Comment 4: Page 4, the experiment of sample #1. ‘However, we have not reached zero resistance due to the incomplete reaction.’ Is there any experimental evidence to support the ‘incomplete reaction’?

Reply: *Many thanks for the reviewer’s questions. The T_c of $Y_{0.5}Ce_{0.5}$ alloy is 4 K at 128 GPa (Supplementary Fig. S4). Thus, the “incomplete reaction” in sample1 is related to the residual $Y_{0.5}Ce_{0.5}$ alloy. Similar situation can be manifested in sample 6 (Supplementary Fig. S6). The XRD patterns of sample 6 also supports the residual $Y_{0.5}Ce_{0.5}$ alloy (Supplementary Fig. S12).*

Comment 5: Page 6. ‘Similar results have been discussed in LaHx and LaH6–7 with the same crystal structure 5,23.’ What this sentence tries to convey is not clear.

Reply: *Many thanks for the reviewer’s questions. This sentence means that the phase of C2/m is reported in LaHx and LaH6-7 in literatures. In the revised manuscript, we have eliminated the unclear sentence.*

Comment 6: Page 6. The lattice constants of $Y_{0.5}Ce_{0.5}H_9$ are indicated. The angle of the C2/m phase is missing.

Reply: *Many thanks for the reviewer’s questions. The experimental lattice parameters, volumes, and hydrogen concentration for sample 1 and sample are summarized in Table S2 in the revised manuscript. The revised manuscript mainly focuses on the P63/mmc phase with high- T_c superconductivity.*

Comment 7: Organization of the manuscript. This referee recommends reorganizing the manuscript for readability and avoiding confusion. More specifically, it would be better to switch the position of samples #4 and #1. When

this reviewer saw Fig. 1, particularly the Raman scattering spectrum, this reviewer wondered why the authors could identify the marked (*) asterisk from the sample. Also, similar questions arise, such as 'Is the peak at 125 cm⁻¹ a peak? Isn't it a noise?' The results of #4 are cleaner and more convincing that the new peak appears after heating.

Reply: Many thanks for the reviewer's comments. We have organized the manuscript with substantial revisions. The cleaner and more convincing Raman spectra are shown in Fig. 1 in the revised manuscript.

Comment 8: Experimental information. The present manuscript tries to provide every single detail of experimental conditions. This reviewer personally truly likes such kind of sincere attitude. However, the manuscript lacks the temperature during laser heating. If the authors did not measure the temperature, it should be mentioned. It also would be a good idea to write the machine model name of the Raman scattering measurement system. It is a good reference for readers.

Reply: Many thanks for the reviewer's comments. The laser heating temperature is about 2000 K (supplementary Table S1). The Raman spectra were measured by an in-house Raman system with a 1300 pixel charge-coupled device designed by Princeton Instrument.

Comment 9: Raman scattering peak analysis. The multiple peak fitting shown in Fig. 1b lower panel looks strange. S(1) peak is symmetric. On the other hand, S(0) is asymmetric. Besides, S(0) seems to include a background. E2g is not fitted. How did the authors assume the peak shapes in their fitting? How was a background handled? Did the authors conduct the multiple peak fitting using all four peaks?

Reply: Many thanks for the reviewer's questions. The Raman peaks were fitted by the Lorentzian-shape function. A linear background was subtracted before fitting the peaks. The four peaks were fitted by multiple peak fitting (Fig. 1c) in the revised manuscript.

Minor points. Page 6. 'Thus, the successfully synthesized Y_{0.5}Ce_{0.5}Y₉ hope to advance the search of alloy hydrides with high-T_c at relatively moderate pressure.' The underlined part should be like 'suggests the hope.'; Page 8. 'The Y_{0.5}Ce_{0.5} pieces were cut into 15-' were -> were; Fig. 1a, 2a, 3a, S1. The pictures placed side-by-side should be aligned in the same orientation to make it easier to see the samples' shape changes; The title of Fig. S1 must be wrong. Run 2 did not observe the superconductivity due to the technical problem.

Reply: Many thanks for the reviewer's comments. All the mentioned minor errors

have been corrected in the revised manuscript.

Response to the second reviewer's comments on NCOMMS-22-29561

Comment 1: Raman data are shown on the reacted phase, but these show little beyond the presence of H₂, evolved from the AB. A very weak, low wavenumber feature seen around 120 cm⁻¹ is attributed to a phonon mode of Y:Ce super hydride. However, this yields little information on the nature or composition of the hydride: there is certainly no clear difference between “incomplete” reacted sample #S1 and the fully reacted #S3 and #S4.

Reply: Many thanks for the reviewer's comments. We appreciate the reviewers for pointing out the lack of detailed description for the Raman scattering measurements. Raman spectra for the synthesized Y_{0.5}Ce_{0.5} hydrides were collected at the apparent-changing part in the sample chamber. Thus, there is certainly no clear difference between different samples. In the revised manuscript, the cleaner Raman spectra are shown in Fig. 1. In addition, the Raman spectra mainly expressed the occurrence of chemical reaction in sample chamber after laser heating.

Comment 2: The authors observe a fairly clear optical change in their sample, which appears to grow larger after the laser-induced reaction in at least two of the sample for which images are shown (#S4 shows little change). The increased size of the sample is interpreted as hydride formation; however, this is very qualitative. The same effect could also result from gasket thinning causing the sample to thin, thus increasing its area without increasing its volume.

Reply: Many thanks for the reviewer's comments. Compared with the pressure before heating, the pressure changes little after heating. It seems reasonable to compare the change of sample area at the same pressure level. In the revised manuscript, we mainly emphasize the shape and color changes of the sample before and after heating.

Comment 3: Unfortunately, I have some serious concerns with the analysis of the diffraction data. Firstly, it's a shame that diffraction data are only shown for sample #S1, which is only partly reacted and does not show a full SC transition. The partial reaction presumably means we should expect unreacted Y:Ce metal to be potentially present. In addition, although they it will scatter rather weakly, there may be some c-BN present as a bi-product of the AB de-hydrogenation. However, neither possibility seems to have been considered in the analysis presented.

Reply: Many thanks for the reviewer's comments. Due to limitation of the diffraction angle for Cell-1(sample1), it is not possible to collect the XRD patterns for c-BN, Pt metal, and Y_{0.5}Ce_{0.5} metal. Thus, another resistance-cell (Cell-6) with larger

diffraction angles was prepared to perform the XRD measurements. The $Y_{0.5}Ce_{0.5}$ hydride phases ($P6_3/mmc$, $I4/mmm$ and $c2/m$), PtH ($P6_3/mmc$), and $Y_{0.5}Ce_{0.5}$ alloy are collected in sample 6 (supplementary Fig. S10-12).

Comment 4: The sample is analyzed as being comprised of a mixture of three hydride phases with space groups: $P6_3/mmc$ (hcp), $I4/mmm$ (tetragonal) and $c2/m$ (monoclinic). This already provides a lot of flexibility to fit the observed diffraction pattern but clearly fails to describe the observations between 15-16 degrees: where only a single peak (from the monoclinic phase) is predicted, two peaks are clearly observed (Fig A blue arrows).

In addition, the largest diffraction feature, seen at 14 degrees, to my eye has more structure (at least 3 peaks) than the two available peaks (one tetragonal one monoclinic) allow. This structure is hidden by the chosen x-axis scaling, but is revealed in the difference curve. I have tried to highlight by changing the aspect ratio. The inflections are quite clearly visible (FigB blue arrows) and I think would be obvious if a greater zoom-range were used in this region. Lastly, I had a similar observation around 17 degrees, where there is a clear, unindexed shoulder (C blue arrow) on the left-hand side of the peak. The conclusion I draw from these observations is that there are issues with the indexing (i.e. allocation of observed intensity to model peaks) and most likely additional phases in the pattern that are not correctly treated.

Reply: Many thanks for the reviewer's comments. Considering the possible inhomogeneous superconducting phases in the electrical cells, we have collected the synchrotron XRD patterns with $5 \mu\text{m}$ steps across the culet. Due to the complex composition in the sample chamber, it is difficult to find a region with pure phase. The XRD peaks of PtH- $P6_3/mmc$ (~ 14 - 16 , ~ 17 degrees) are always searched in the sample, which are induced by the chemical reaction of Pt leads and hydrogen. The represented XRD patterns including $Y_{0.5}Ce_{0.5}H_9$ - $P6_3/mmc$ and PtH- $P6_3/mmc$ are

shown in Fig. 4 in the revised manuscript. The refined parameters of $Y_{0.5}Ce_{0.5}H_9$ - $P6_3/mmc$ with the Rietveld method are given in supplementary Table S2.

Comment 5: A second technical issue is observed with the peak width for the tetragonal phase. The Le Bail refinement is clearly strongly steered towards fitting the isolated peak just above 12 degrees. This peak is well fitted, but the other peaks due to the same phase will be constrained to have similar width. This clearly misfits the data at 18 degrees (Fig C red arrow), so I am left wondering about the allocation of intensity in the main peak at 14 degrees.

Reply: Many thanks for the reviewer's comments. Due to the complex composition in the sample chamber, it is difficult to refine the XRD patterns very well, especially the peaks at large angles. To solve this issue, the XRD patterns (Fig. 4) only including $P6_3/mmc$ - $Y_{0.5}Ce_{0.5}H_9$ and PtH- $P6_3/mmc$ phases were studied in revised manuscript. As mentioned above, the XRD peaks of PtH- $P6_3/mmc$ cannot be ruled out in the sample, due to the chemical reaction of Pt leads and hydrogen.

Comment 6: A final criticism I had is with the inclusion of the crystal structures as insets to Figure 4a. These imply that a definitive structure is known when, in fact, the observations only yield heavy atom positions and an inferred (and, likely inaccurate, on the basis of above comments) estimate of hydrogenation from volumetric measurements. So, how are the hydrogen sites known? Whether these are taken from calculation, from citation, inferred, or just guessed is not clear as the caption merely states "inset shows the crystal structure". There is also inaccuracy in the color coding of the Ce and Y atoms. These are shown to separately occupy equivalent atomic sites in the crystal structure, making them inequivalent. Thus, the figures are not consistent with the space groups given. In reality, these species are substitutionally disordered over the same crystallographic site in a way that the crystal structure images misrepresent.

Reply: Many thanks for the reviewer's comments. The hydrogen content of the synthesized $Y_{0.5}Ce_{0.5}$ alloy hydrides was determined as 8.3-8.8 through the cell unit volume. The calculated hydrogen concentration has a small deviation with the ideal value of 9, because of the direct use of elemental phases for the evaluation of volumes. The hydrogen sites for $P6_3/mmc$ - $Y_{0.5}Ce_{0.5}H_9$ are referring from the literature. Combined with recently discovered structure of $P6_3/mmc$ - CeH_9 , our diffraction experiments show that 50% of the Ce atoms are replaced by Y, and thus forming a unique ternary alloy superhydride $P6_3/mmc$ - $Y_{0.5}Ce_{0.5}H_9$. The color coding of the Ce and Y atoms in the crystal structure of Fig. 4b has been corrected in the revised manuscript.

Response to the third reviewer's comments on NCOMMS-22-29561

Comments: I do realize that any measurements in the megabar pressure range are extremely difficult and require a lot of effort. This said, the presented transport data do not convince the reader that there is bulk superconductivity in the samples (in principle, one can have zero resistance with filamentary superconductivity associated with some minority phase). The author did not report any attempts at further measurements that would make this claim stronger: resistance in a magnetic field, I-V curves, ac susceptibility, dc susceptibility, etc...

Reply: *Many thanks for the reviewer's comments. To confirm the superconducting state, we performed the electrical transport measurements under external magnetic field for sample 7 at 166 GPa (Fig. 3) in the revised manuscript.*

REVIEWER COMMENTS

Reviewer #1 (Remarks to the Author):

The 2nd review report for manuscript NCOMMS-22-29561A entitled 'Synthesis and superconductivity in yttrium-cerium hydrides at moderate pressures' authored by Liu-Cheng Chen et al.

The key achievements of this study are;

1. The successful synthesis of ternary hydrides $Y_{0.5}Ce_{0.5}H_9$ under pressure and the observation of the superconductivity with superconducting transition temperatures (T_c) ranging from 97 to 140 K by electrical resistance measurements.
2. The observed T_c are higher than the previously reported CeH_9 and lower than YH_9 , meaning the ternary hydride can maintain the high- T_c superconductivity.
3. The X-ray diffraction measurements observed three structural phases $P63/mmc$, $I4/mmm$, $C2/m$). The theoretical calculation suggests that the $P63/mmc$ phase is the high- T_c superconducting phase.

Summary of the review

This study's achievements listed above are highly evaluated to yield important progress in the efforts to obtain superhydride superconductors at pressures as low as possible. However, on the other hand, the author's claim that $Y_{0.5}Ce_{0.5}H_9$ can be synthesized and stabilized at moderate pressure lower than that required for YH_x ($x = 4-9$) is not convincing. Furthermore, this reviewer must point out that deducing such a claim is impossible, considering the available experimental data for YH_x ($x = 4-9$) and CeH_9 . The mismatch between the study's design and the purpose of the study remains even after the first revision. This reviewer thinks the claim 'the synthesis at moderate pressures' should be dropped. The previous studies of YH_x and CeH_x have revealed or tried to reveal at least, which structural phase possesses how much of T_c . Some did that experimentally, and some were theoretically. Considering those facts, the present study needs further discussion about the structural phases and their T_c to be considered for publication. In addition, this manuscript still contains a couple of conflicts and unrigorous discussions listed below. Although the improvement of the manuscript, including the additional experiments, is impressive, this reviewer still does not come to recommend the study for publication in Nature Communications.

Comments, questions, and criticisms

1. Claim of 'synthesis at modest pressure' is misleading and mismatches the study's design.

This reviewer pointed out the same in the previous manuscript. The authors addressed the criticisms by comparing the reported synthesis conditions and stability region of YH_4 , YH_6 , YH_9 , and CeH_9 (summarised in the Supplemental information Table S4.) However, as far as this reviewer finds, the use of the data in Table S4 is not accurate. Among the indicated references, including this study, none have investigated the minimum pressure required for synthesizing YH_4 , YH_6 , YH_9 , CeH_9 , and $Y_{0.5}Ce_{0.5}H_9$. Ref. 4 studied the decomposition pressure of $P63/mmc$ - YH_9 and $Im-3m$ - YD_6 . Other studies did not reveal the decomposition pressures in their studies. Besides, the indicated lower stability limit (160 GPa) of $Im-3m$ - YH_6 seems to conflict with the value reported (135 GPa) in Ref. 4. (<- Please clarify.) This study did not observe the decomposition of $Y_{0.5}Ce_{0.5}H_9$ upon decompression. As a whole, and in principle, concluding decisively that the synthetic pressure and minimum stable pressure of $Y_{0.5}Ce_{0.5}H_9$ are much lower than YH_x for $x \geq 4$ is impossible. If it is necessary to discuss the stability region and the required minimum pressure, the authors should write clearly what the shown data (synthesis pressure and temperature, stability region) indicate exactly. Specifically, for example, 143 GPa for the synthetic pressure of $I4/mmm$ - YH_4 is the lowest pressure tested in Ref. 2. But, it is not the required minimum pressure for the material.

2. Conflicts in T_c determination

The main text writes that the T_c is defined as 90% of the normal-state resistance. However, in Fig. 2a and 3a, the authors seem to determine it at the crosspoint of the two extrapolated lines from a superconducting and normal state $R </R>$ vs. T . Which is correct? If one reads the temperature at 90% of normal-state resistance, the T_c at 166 GPa for the Cell-7 is 130 K, not 138 K. This correction makes Fig. 2a look different from the current one. Also, in Fig. S6, the definition seems different from both Fig. 2a and 3b. The T_c determination must be conducted consistently. Figures need correction. Likewise, the discussions and estimation of the upper-critical field and coherent length must be revised.

3. Pressure dependence of T_c

Fig. 2b and plotting the T_c as a function of pressure, especially showing the guide for eyes (grey dotted line), is not simple.

- Three phases exist simultaneously in a sample.
- Although the authors' theoretical calculation suggests the P63/mmc phase is responsible for the highest T_c , the other two phases remain the potential contributors to the observed superconducting transition. In fact, the R vs. T curves below the T_c show multiple steps, indicating multiple phases exhibit superconducting transitions. In addition, the hydrogen content is also a factor in determining the T_c in the same structural phases. Those facts suggest that the T_c vs. P in Fig. 2b likely consists of several T_c vs. P curves for different phases.
- The authors also do not know what hydrogen content enables the highest T_c . The pressure dependence of the T_c for each phase is also unknown.

Obtaining superconducting hydrides at lower pressure, with as much higher T_c , is the ultimate purpose of the present study. The previous studies of YH_x and CeH_x revealed/suggested which structural phases and hydrogen contents possessed how much T_c . Considering the purpose, the experimental facts of the present study, and the achievements of the previous studies, the authors need discussions about the T_c s for the three phases (P63/mmc, I4/mmm, C2/m) potentially have. Discussions about how the hydrogen content affects the T_c and the pressure dependence of the T_c of each phase are desirable. Can the theoretical calculation be extended to the other two phases? May analyzing the R vs. T curves and the steps below T_c carefully give some thought?

4. Rietveld refinement, Fig. 4a: The fitting results (R_p and R_{wp} -values) must be shown.

5. Unclear wording: Page 7, line 4 from the bottom, 'The calculated hydrogen concentration has a small deviation with the ideal value of 9, because of the direct use of elemental phases for the evaluation of volumes.' What does this sentence mean? Is the value 8.3-8.8 inaccurate because of the hydrogen content estimation method?

• Manuscript organization

1. Page 3, the last paragraph It would be helpful to readers to describe why the authors chose $Y_{0.5}Ce_{0.5}$ superhydrides as their target material. What strategy/principle did the authors employ in selecting materials? Did they obtain some suggestions from theoretical studies?

2. Page 5, line 5 'For comparison, the temperature-dependent resistance of $Y_{0.5}Ce_{0.5}$ alloy with $T_c \sim 4$ K was measured before heating (Supplementary Fig. S4).' What is the purpose of the comparison? Writing more specifically would be better.

3. Page 5, 2nd paragraph The 'sample 7' should be written Cell-7 per the notation scheme in this manuscript.

4. Page 5, 2nd paragraph The utilized calibration curve for a diamond Raman pressure determination method must be in the references list.

5. Correction in references: Reference 7 in the Supplemental information, 'Takahiro, M., Masahiro, H., Keiji, K., Naohisa, Hi., Yasuo, O., Shigeo, S., Kazushi, T. & Katsuya, S. Superconductivity of platinum hydride. Phys. Rev. B 99, 144511 (2019).' All family and given names are inverted. ex.)Takahiro, M. -> Matsuoka, T.

● **Minor corrections**

1. Page 3, Line 2: 'It is clear that the experimentally obtained YH_x superconductors are only stable at extreme pressures (above ~ 155 GPa).' Please provide reference materials.

2. Page 3, 2nd paragraph: 'Superconductivity in lanthanum-cerium ternary superhydrides with $T_c \sim 176$ K was discovered at the moderate pressures 23,24. 'Theoretical calculations demonstrated that a series of ternary superhydrides can hold high T_c 's at relatively low stabilized pressure.' ' Readers would want to know the pressure value here.

I've considered the changes made by the authors and think they have significantly improved the manuscript. In particular, the different diffraction dataset they now show looks to be cleaner than the data used previously and the integrated pattern more convincingly show two peaks indexed on the hcp lattice (the 102 reflection is at $\sim 17.2^\circ$ is indexed but doesn't really have intensity above the noise level). Furthermore, diffraction patterns for other pressures and locations in the cell are given in the supplementary material. On balance, I think it's credible to say that the indexed P 63/mmc YCeHx phase is present on the basis of the diffraction data.

On this basis, I can recommend publication, although I still felt that some important information should be added:

- 1) The 2d x-ray image data are reported as being "converted to one dimensional diffraction data with Dioptas." More detail should be given on exactly how the conversion was done. Specifically, was a background subtracted and what parameters were used to do this? The importance of this is that knowledge of background level allows an assessment of expected noise due to it. This could be shown in a different way by indicating the counting error on the plots showing the fits: this is commonly done either as error bars on the data, or by scaling the residual by the error. I think this is important for very low signal-to-background measurements such as these, so a reader can make some judgement on which is which.
- 2) On page 11, the statement is made that the diffraction data were 'analyzed...[using] Le Bail or Rietveld' methods. The caption for Figure 4a indicates that the calculated curve is a Rietveld fit. Rietveld has a pre-requisite that the beam must illuminate a good statistical average of crystallite orientations. In this case, the powder lines will be continuous and smooth around the complete Debye ring. This is not observed here, where the diffraction lines consist of only a handful of isolated spots. Consequently, Rietveld analysis is not appropriate and, if it was used, the parameters that were refined, and their refined values should be stated. On the basis of the quality of diffraction data I would not expect these to be physically meaningful. In addition to structural parameters, details such as background type and number of terms as well as peak parameters (such as peak profile used and refined profile parameters) should normally be given in supplementary material.
- 3) In their rebuttal to my previous comment 6, the authors say "our diffraction experiments show that 50% of the Ce atoms are replaced by Y". Related to my point above, this actually cannot be shown from their diffraction data, as the powder lines do not have valid relative intensities due to poor statistical sampling of crystallite orientations. Consequently, site occupation factors cannot be determined from the diffraction data shown.
- 4) The cake view of the pattern shows the clear presence of a peak that does not show up in the integrated pattern and is not related to either YCeHx or PtH:

is there an identification of this feature?

Reviewer #3 (Remarks to the Author):

Of all resistance data presented in the manuscript, apparently only two cells, cell 3 and cell 7 present instrumental zero resistance. Of these two the transition width for cell 7 is over 100 K. The rest of the cells have incomplete resistive transitions. Even if I decide to agree with the authors that they observe superconductivity (note, there is an ongoing, heated at times, discussion, involving multiple experimental techniques, if any of superhydrides is a superconductor), it seems to be filamentary, not bulk. In my opinion this manuscript does not present any data that allow to associate the alleged superconductivity with any particular phase of the multiphase material present in DAC. This said I cannot recommend this manuscript for publication in Nature Communications.

Additional comment is related to band structure calculations: what valence of Ce was assumed and what experimental evidence this assumption was based on? If there is any magnetic moment on Ce then there is an additional, Abrikosov - Gorkov mechanism of T_c suppression, that is not included in the presented calculations.

Summary of change on manuscript # NCOMMS-22-29561A

We have revised the manuscript in accordance with all the comments and suggestions from three reviewers. The detailed changes are summarized as follows:

1. We have added the R-T curves of Cell-6 (named Cell-7 in the previous version) at higher pressures (166-200 GPa) (Supplementary Fig. S7).
2. We have added one more DAC (Cell-7) with sharp superconducting transition and clear zero resistance at the pressures of 124-155 GPa (supplementary Fig. S8).
3. We have added the temperature dependence of the resistance at applied magnetic fields for Cell-7 at 124 GPa (Fig. 3).
4. We have provided more details about the structural refinements in supplementary Table. S2.
5. More discussions about the Y/Ce ratio in the hydride have been added.
6. The figures in the main text have been adjusted after adding the new experimental data.
7. The order of the figures in supplementary information has been adjusted.
8. All the minor errors pointed by the reviewers have been corrected.
9. The references have been updated and added accordingly.

Response to the first reviewer's comments on NCOMMS-22-29561A

We thank the reviewer for his/her evaluation of the three key achievements of the study of our work. We highly appreciate the reviewer's encouragement for our work in yielding important progress in the efforts to obtain superhydride superconductors at pressures as low as possible. The reviewer judged "Although the improvement of the manuscript, including the additional experiments, is impressive, this reviewer still does not come to recommend the study for publication in Nature Communications." Therefore, the reviewer suggested us to make some additional improvements from the technical points in order to deliver the best quality of our work for the publication.

We took a very careful look at all the suggestions from the reviewer. The major concern is the enhancement of the superconducting transition temperature of $\text{Ce}_{0.5}\text{Y}_{0.5}\text{H}_9$ at modest pressure compared to CeH_9 and YH_9 . The reviewer thought that our claim is not convincing. He/She hopes to see a systematic comparison of the experiments to support this finding. After having fully adopted this nice suggestion, we revised the manuscript again from every aspect. Now the detailed response for this concern as well as other technical points will be given as follows.

Point I: "Claim of 'synthesis at modest pressure' is misleading and mismatches the study's design: *This reviewer pointed out the same in the previous manuscript. The authors addressed the criticisms by comparing the reported synthesis conditions and stability region of YH_4 , YH_6 , YH_9 , and CeH_9 (summarized in the Supplemental information Table S4.) However, as far as this reviewer finds, the use of the data in Table S4 is not accurate. Among the indicated references, including this study, none have investigated the minimum pressure required for synthesizing YH_4 , YH_6 , YH_9 , CeH_9 , and $\text{Y}_{0.5}\text{Ce}_{0.5}\text{H}_9$. Ref. 4 studied the decomposition pressure of $P63/mmc\text{-YH}_9$ and $Im\text{-}3m\text{-YD}_6$. Other studies did not reveal the decomposition pressures in their studies. Besides, the indicated lower stability limit (160 GPa) of $Im\text{-}3m\text{-YH}_6$ seems to conflict with the value reported (135 GPa) in Ref. 4. (<- Please clarify.) This study did not observe the decomposition of $\text{Y}_{0.5}\text{Ce}_{0.5}\text{H}_9$ upon decompression. As a whole, and in principle, concluding decisively that the synthetic pressure and minimum stable pressure of $\text{Y}_{0.5}\text{Ce}_{0.5}\text{H}_9$ are much lower than YH_x for $x \geq 4$ is impossible. If it is necessary to discuss the stability region and the required minimum pressure, the authors should write clearly what the shown data (synthesis pressure and temperature, stability region) indicate exactly. Specifically, for example, 143 GPa for the synthetic pressure of $I4/mmm\text{-YH}_4$ is the lowest pressure tested in Ref. 2. But, it is not the required minimum pressure for the material."*

Reply: We thank the reviewer for this concern and for the careful check of the data in the literature. Indeed, the comparison of the obtained T_c values is helpful to judge whether the claim made in this work is foundational or not. For the reliable and physically reasonable comparison, we concentrate ourselves on the data points in hydrides XH_9 ($X=Y, Ce,$ and $Ce_{0.5}Y_{0.5}$) with the same hydrogen content and crystal structure. The following figure summarizes our data points in the present work together with those of YH_9 (Ref. [10,12]) and CeH_9 (Ref. [17]), respectively. The results support our claim. However, this claim is a tiny issue and should not affect the three achievements as recognized by the reviewer. If the reviewer could not be satisfied to this claim, we would like to change the words of “modest pressures” into “high pressures” in the title of this manuscript.

Figure: Pressure dependence of T_c for YH_9 [Refs.10,12], CeH_9 [Ref.17], and $Y_{0.5}Ce_{0.5}H_9$ from the present work.

Point II: “Conflicts in T_c determination: The main text writes that the T_c is defined as 90% of the normal-state resistance. However, in Fig. 2a and 3a, the authors seem to determine it at the crosspoint of the two extrapolated lines from a superconducting and normal state $R</R>$ vs. T . Which is correct? If one reads the temperature at 90% of normal-state resistance, the T_c at 166 GPa for the Cell-7 is 130 K, not 138 K. This correction makes Fig. 2a look different from the current one. Also, in Fig. S6, the definition seems different from both Fig. 2a and 3b. The T_c determination must be

conducted consistently. Figures need correction. Likewise, the discussions and estimation of the upper-critical field and coherent length must be revised.”

Reply: We have unified the definition of T_c with the crossing point of the two extrapolated lines from the superconducting and normal state. The conflicts in T_c determination have been figured out. The whole manuscript has been revised accordingly.

Point III 1: *“Pressure dependence of T_c : Fig. 2b and plotting the T_c as a function of pressure, especially showing the guide for eyes (grey dotted line), is not simple. Three phases exist simultaneously in a sample. Although the authors’ theoretical calculation suggests the $P63/mmc$ phase is responsible for the highest T_c , the other two phases remain the potential contributors to the observed superconducting transition. In fact, the R vs. T curves below the T_c show multiple steps, indicating multiple phases exhibit superconducting transitions. In addition, the hydrogen content is also a factor in determining the T_c in the same structural phases. Those facts suggest that the T_c vs. P in Fig. 2b likely consists of several T_c vs. P curves for different phases.”*

Reply: We thank the reviewer for pointing out the difficulty in identifying phase and its superconducting properties. The x-ray diffraction data suggests that $P63/mmc$ is the primary phase and the other two are the secondary ones. The obtained phonon spectra are consistent with the predicted phonon peak derived from the calculations based on the primary phase. The observed T_c at the relevant pressure is in fair agreement with the calculated results based on the structure of the primary phase. In addition, the hydrides with the two ending elements possess the obtained $P63/mmc$ structure to holding the highest T_c as their alloy hydride. All these facts together enable us to believe that the observed superconducting character is mainly from the primary phase, *i.e.*, $P63/mmc$ in this work. This is a general criteria for determining the superconductivity in superhydrides in the literature. In fact, we adopted this standard in our recent works, one for $Ce_{0.5}La_{0.5}H_{10}$ (Ref. 33) and the other for CeH_9 (ref. 18) in collaboration with Korean scientists.

Point III 2: *“The authors also do not know what hydrogen content enables the highest T_c . The pressure dependence of the T_c for each phase is also unknown.”*

Reply: We agree with the reviewer. Yes, we do not know what hydrogen content enables highest T_c and the pressure dependence of T_c of each phase. The observed T_c at high superconducting transition is assumed from the $P63/mmc$ phase of $Ce_{0.5}Y_{0.5}H_9$.

Point III 3: “Obtaining superconducting hydrides at lower pressure, with as much higher T_c , is the ultimate purpose of the present study. The previous studies of YH_x and CeH_x revealed/suggested which structural phases and hydrogen contents possessed how much T_c . Considering the purpose, the experimental facts of the present study, and the achievements of the previous studies, the authors need discussions about the T_{cs} for the three phases ($P63/mmc$, $I4/mmm$, $C2/m$) potentially have. Discussions about how the hydrogen content affects the T_c and the pressure dependence of the T_c of each phase are desirable. Can the theoretical calculation be extended to the other two phases? May analyzing the R vs. T curves and the steps below T_c carefully give some thought?”

Reply: We thank the reviewer for the suggestion in the identification of each phase of the synthesized samples. The discussions about how the hydrogen content affects T_c and the pressure dependence of T_c of each phase are indeed valuable from both the experimental and theoretical viewpoints. However, the obtained temperature-dependent resistance curves do not exhibit clear signature of the superconducting transitions except the one at higher temperature defined as T_c . The multiple steps may indicate the nonhomogeneous character of the phase due to the pressure gradient within the diamond anvil cell. Therefore, we have to focus on the highest superconducting transition in the resistance curves and believe it to be associated with the $P63/mmc$ phase. This consideration is consistent with the primary phase character from the x-ray diffraction data, Raman spectra, and calculations. For future investigations, the synthesis of single phase of $I4/mmm$ and $C2/m$ will be necessary to establish the foundation of superconductivity of them as well as the pressure effect on it. This is not the focus of the current study.

Point IV: “Rietveld refinement, Fig. 4a: The fitting results (R_P and R_{wp} -values) must be shown.”

Reply: The fitting results have been given in Supplementary Table S2 of the revised manuscript.

Point V: “Unclear wording: Page 7, line 4 from the bottom, ‘The calculated hydrogen concentration has a small deviation with the ideal value of 9, because of the direct use of elemental phases for the evaluation of volumes.’ What does this sentence mean? Is the value 8.3-8.8 inaccurate because of the hydrogen content estimation method?”

Reply: In ideal conditions, the atoms volumes of Y and Ce at target pressures should be used for obtaining the hydrogen content. However, one cannot obtain the atoms volumes under pressure in the reality. As an alternative, the elemental phases for the evaluation of volumes are always used. This will lead to the small deviation between the ideal value and experimental one.

Point VI 1: *“Manuscript organization, Page 3, the last paragraph It would be helpful to readers to describe why the authors chose $Y_{0.5}Ce_{0.5}$ superhydrides as their target material. What strategy/principle did the authors employ in selecting materials? Did they obtain some suggestions from theoretical studies?”*

Reply: In the introduction of the manuscript, we have provided the clue why we choose $Y_{0.5}Ce_{0.5}$ alloy and synthesize such alloy hydrides with the combined advantages of the low synthesized pressure in CeH_9 and high T_c in YH_9 .

Point VI 2: *“Manuscript organization, Page 5, line 5 ‘For comparison, the temperature-dependent resistance of $Y_{0.5}Ce_{0.5}$ alloy with $T_c \sim 4$ K was measured before heating (Supplementary Fig. S4).’ What is the purpose of the comparison? Writing more specifically would be better.”*

Reply: The comparison of T_c values before and after heating hopes to demonstrate that the observed high- T_c superconductivity is not from $Y_{0.5}Ce_{0.5}$ alloy rather from the alloy-based hydrides. We have added this point in the revised manuscript.

Point VI 3: *“Manuscript organization, Page 5, 2nd paragraph The ‘sample 7’ should be written Cell-7 per the notation scheme in this manuscript.”*

Reply: Many thanks to the reviewer for this point. We have changed the label for the ‘sample 7’ as ‘Cell-7’ in the revised manuscript.

Point VI 4: *“Manuscript organization, Page 5, 2nd paragraph The utilized calibration curve for a diamond Raman pressure determination method must be in the references list.”*

Reply: The reference has been provided in the revised manuscript.

Point VI 5: *“Manuscript organization, Correction in references: Reference 7 in the Supplemental information, ‘Takahiro, M., Masahiro, H., Keiji, K., Naohisa, Hi., Yasuo, O., Shigeo, S., Kazushi, T. & Katsuya, S. Superconductivity of platinum hydride. Phys. Rev. B 99, 144511 (2019).’ All family and given names are inverted. ex.)Takahiro, M. -> Matsuoka, T.”*

Reply: The errors have been corrected in the revised manuscript.

Point VII 1: *“Minor corrections, Page 3, Line 2: ‘It is clear that the experimentally obtained YHx superconductors are only stable at extreme pressures (above ~155 GPa).’ Please provide reference materials.”*

Reply: The refs. [9-12] have been provided in the revised manuscript.

Point VII 2: *“Minor corrections, Page 3, 2nd paragraph: ‘Superconductivity in lanthanum-cerium ternary superhydrides with $T_c \sim 176$ K was discovered at the moderate pressures 23,24. ‘Theoretical calculations demonstrated that a series of ternary superhydrides can hold high T_c ’s at relatively low stabilized pressure.’ ‘Readers would want to know the pressure value here.’”*

Reply: The pressure values have been added in the revised manuscript.

In summary, we have addressed all the points from the reviewer in the revision of the manuscript. We hope the reviewer safely remove his/her concerns for these technical points and recommend the current version of the manuscript for the publication in *Nature Communications*.

Response to the second reviewer's comments on NCOMMS-22-29561A

We would like to express our sincere thanks to the second reviewer for his/her evaluation of our work with the significant improvement. We have benefited a lot from the reviewer's professional review and excellent suggestions on the analysis of our structural data. The reviewer is highly appreciated for his/her recommendation for the publication of this work in *Nature Communications*.

In the following, we would like to provide the detailed response to the reviewer's additional suggestions for us to improve the presentation and quality of our work for the publication.

Suggestion I: *“The 2d x-ray image data are reported as being ‘converted to one dimensional diffraction data with Dioptas.’ More detail should be given on exactly how the conversion was done. Specifically, was a background subtracted and what parameters were used to do this? The importance of this is that knowledge of background level allows an assessment of expected noise due to it. This could be shown in a different way by indicating the counting error on the plots showing the fits: this is commonly done either as error bars on the data, or by scaling the residual by the error. I think this is important for very low signal-to-background measurements such as these, so a reader can make some judgement on which is which.”*

Reply: Many thanks to the reviewer's for this suggestion. The background has been subtracted automatically with the default parameters when converting the XRD patterns by using Dioptas (supplementary Table S2). We have provided the details in the reduction of the data in the revision.

Suggestion II: *“On page 11, the statement is made that the diffraction data were ‘analyzed...[using] Le Bail or Rietveld’ methods. The caption for Figure 4a indicates that the calculated curve is a Rietveld fit. Rietveld has a pre-requisite that the beam must illuminate a good statistical average of crystallite orientations. In this case, the powder lines will be continuous and smooth around the complete Debye ring. This is not observed here, where the diffraction lines consist of only a handful of isolated spots. Consequently, Rietveld analysis is not appropriate and, if it was used, the parameters that were refined, and their refined values should be stated. On the basis of the quality of diffraction data I would not expect these to be physically meaningful. In addition to structural parameters, details such as background type and number of terms as well as peak parameters (such as peak profile used and refined profile parameters) should normally be given in supplementary material.”*

Reply: Many thanks to the reviewer again for this nice suggestion. It is a big challenge to analyze the diffraction data with good statistical average of crystallite orientations for the superhydrides at such high pressures by using diamond anvil cells with small opening. The isolated points caused by single-crystal-like diffraction are often observed for the products after being heated at high temperature and high pressure. For such diffraction patterns, it is indeed very hard to draw the reliable structural information from Rietveld' method. However, we would like to make effort to provide such information by Rietveld analysis rather than Le Bail fitting. The detailed parameters are shown in supplementary Table S2 in the revised manuscript.

Suggestion III: "In their rebuttal to my previous comment 6, the authors say "our diffraction experiments show that 50% of the Ce atoms are replaced by Y". Related to my point above, this actually cannot be shown from their diffraction data, as the powder lines do not have valid relative intensities due to poor statistical sampling of crystallite orientations. Consequently, site occupation factors cannot be determined from the diffraction data shown."

Reply: The stoichiometric ratio of prodromic Y-Ce alloy is 1:1 as expected from the same stoichiometric ratio of Y and Ce in the synthesized hydrides. In the XRD patterns, we failed to detect any peaks corresponding to those belonging to either elemental Y or Ce besides the peaks from the synthesized hydrides. This suggests that the precursor of $Y_{0.5}Ce_{0.5}$ alloy completely reacts with hydrogen with the molar ratio of $\sim 1:1$ for the formation of hydrides. As a result, Y/Ce ratio in the formed hydrides should be 50% to 50%. The analysis of the Y/Ce ratio is detailed in revised manuscript.

Suggestion IV: "The cake view of the pattern shows the clear presence of a peak that does not show up in the integrated pattern and is not related to either $YCeH_x$ or PtH is there an identification of this feature?"

Reply: Thank the reviewer again for this suggestion. This is a weak signal, which may be deducted as background in the previous version of the manuscript. The XRD patterns were converted again with default parameters by using Dioptas in the revised manuscript. And the background of XRD patterns was subtracted automatically with the default parameters in this process (Figure 4 and supplementary Table S2). That means the background of XRD data was subtracted before the refinement.

We would like to thank the second reviewer again for his/her excellent suggestions. All these additional suggestions have been fully addressed in the revised manuscript.

Response to the third reviewer's comments on NCOMMS-22-29561A

We would like to express our appreciations to the third reviewer for his/her time, attention, and energy in reviewing our manuscript. The reviewer was not fully satisfied to our experiments and calculations. He/she hopes us to present more experimental results to support the bulk superconductivity rather than the filamentary one. Meanwhile, he/she wonders whether the consideration of the magnetic character of Ce could affect the calculated superconducting transition temperature. These considerations are very valuable for us to improve the quality of our work.

After fully having adopted his/her comments and suggestions, we re-synthesized one more sample at high pressures and re-conducted the resistance measurements as a function of temperature with and without the applied magnetic fields. The results clearly show the sharp superconducting transition at the temperature similar to what we observed previously at the almost same pressure(s). The realization of the zero-resistance state as well as the suppression of the superconducting transition upon the magnetic field clearly supports the bulk superconductivity in the studied Ce-Y-alloy hydrides. Together with the newly prepared diamond anvil cell, we have total three cells with the zero-resistance state. Although the more is better, one cell might be enough to provide the evidence for supporting superconductivity. In reality, this is the case for La-H. At that time, M. Somayazulu and his coworkers just observed the zero-resistance state from one of their diamond cells [ref. 3 of our manuscript, *Phys. Rev. Lett.* 122, 027001 (2019)]. They even did not perform the temperate-dependent resistance measurements with magnetic fields. In the late experiments, A. P. Drozdov and his co-worked observed the zero-resistance state in four of their cells [ref. 4 of our manuscript, *Nature* 569, 528-531 (2019)]. For Y-H, Kong *et al.* reported superconductivity up to 243 K under pressure through the observations of the zero-resistance state in three diamond anvil cells [ref. 10 of our manuscript, *Nat. Commun.* 12, 5075 (2021)]. A reasonable comparison of our work with these publications supports our conclusion for the observed bulk superconductivity in the studied Ce-Y-H system. Actually, it is very hard to believe that Ce_{0.5}Y_{0.5}H₉ does not hold bulk superconductivity with the signature similar to the observations in YH₉ and CeH₉.

The important contribution of our work is the observation of higher critical temperature in Ce_{0.5}Y_{0.5}-alloy hydride than Ce-H but at lower pressure than Y-H with the same hydrogen content and crystal structure. This demonstrates the enhancement of the superconducting transition temperature at modest pressure. The study points to a new direction to have high superconducting transition temperature at low pressure or even ambient pressure. We shall appreciate the reviewer very much if his/she could remove the concerns about the bulk superconductivity and recommend our work for publication based on the mentioned significance.

Regarding the additional comment related to band structure calculations, the reviewer asked “what valence of Ce was assumed and what experimental evidence this assumption was based on”. The reviewer wondered whether T_c suppression could happen based on the Abrikosov - Gorkov mechanism if there is any magnetic moment on Ce.

The purpose of the theoretical calculations is to help the identification of the experimentally obtained phase(s). The calculations are thus in the secondary position of the current work. In the calculations, we took the valence of Ce as $5s^26s^25p^65d^14f^1$ and did not include the magnetic moment of Ce. This consideration is probably true because there is no evidence for the occurrence of magnetic moment of Ce at the studied high pressures. For most magnetic element, the magnetic moment could be suppressed or quenched by the applied pressure. In this aspect, Fe may serve an example. The application of pressure can suppress the magnetic moment and then drives it to superconductivity. This could be the reason why the works on Ce-H system from the theoretical groups have not considered the magnetic moment in their calculations [ref. 17 of our manuscript, *Phys. Rev. Lett.* 127, 117001 (2021)].

Our calculations were used to evaluate the structure and phase stability with the comparison of the experimental data from both the x-ray diffraction and Raman scattering measurements. The results support the obtained P_{63}/mmc -Y_{0.5}Ce_{0.5}H₉. The large contribution of the electronic states of Ce-f indicates that the delocalized nature of Ce-f electrons is responsible for the enhanced chemical pre-compression in Y_{0.5}Ce_{0.5}

hydrides. The theoretically obtained T_c of 104-119 K is in fair agreement with our experiments. This suggests that the inclusion of magnetic moment or not for Ce could not change the conclusions in this work.

From the above detailed responses, we hope the reviewer remove his/her concerns regarding the bulk superconductivity established from experiments and the magnetic moment of Ce in the calculations and recommend this manuscript for the publication in *Nature Communications*.

REVIEWER COMMENTS

Reviewer #1 (Remarks to the Author):

This is the 3rd review report for manuscript 380516 entitled 'Synthesis and superconductivity in yttrium-cerium hydrides at moderate pressures.'

After two previous reviews, the authors revised the manuscript to address the criticisms and questions of three reviewers. For this reviewer's comments, the authors' addresses are mostly satisfactory.

This reviewer's main concern was the misleading writing that said the Y-Ce-hydride superconductor is synthesized and stabilized at moderate pressures.

In the current manuscript, that claim was dropped, and the authors focus their discussion on the Tc and structural phases of the synthesized Y-Ce-hydrides, not comparing the stability pressure region with Y-hydrides and Ce-hydrides.

Regarding the title of the manuscript, as stated by the author, this reviewer thinks it is better to change the words 'modest pressures' -> 'high pressures' per the discussion of the manuscript.

Besides, this reviewer suggests including the figure 'Tc vs. Pressure' for CeH_x, YH_x, and Ce-Y-H_x shown in response to this reviewer's comments. This figure certainly helps readers understand the achievement of this study much better than comparing the data on tables or texts.

As a summary, this reviewer thinks the current manuscript can be recommended for publication in Nature Communication. Further review is not necessary for this reviewer.

Reviewer #2 (Remarks to the Author):

Referee 2 Suggestion II: The authors say they "want to *make the effort* to provide such information by Rietveld analysis". However, my point was that Rietveld analysis is not appropriate, because the measured relative diffraction intensities are not sufficiently well sampled. It doesn't matter how much effort is expended if the information is not present in the data due to inadequate statistical sampling of crystallite orientations. The authors do not make any claims in relation to crystal structure that would rely on accurate intensities (e.g. atomic positional parameters), so the use of Rietveld is not only unjustified, but also unnecessary.

Referee 2 Suggestion III: the authors claim their diffraction data confirm stoichiometry because "In the XRD patterns we failed to detect any peaks corresponding to ...elemental Y or Ce". However, there were unindexed and undiscussed peaks in the 2θ pattern they show, which I highlighted in my suggestion IV. How is it possible to know that these peaks do not come from an unidentified phase containing Y or Ce?

Referee 2 Suggestion IV: The authors have used a background subtraction method that incorrectly removes sharp, although weak, diffraction peaks, so these were not considered. This oversight allows for misinterpretation, such as the one I highlight above.

In light of the comments above, I still feel there are areas of the manuscript where conclusions are drawn that exceed the limits of the data obtained.

Reviewer #3 (Remarks to the Author):

I would like to divide my report into two parts:

- comments on the text and SI;
- comments on the Response on Referee Letter

In my opinion, several statements in the text are not formulated with a proper precision and therefore might sound misleading:

- abstract: "The superconductivity is detected with the zero-resistance state at the critical temperature in the range of 97-141 K..." The temperatures cited are the temperatures of the onset of the superconducting transition. Zero resistance state was not achieved in half of the experiments, Zero resistance state in cells 3, 6, 7 in Fig. 2a is achieved between ~ 115 K and ~ 10 K.

- abstract: "The upper critical field towards 0 K at 124 GPa is determined with the large value of 78 T by the electrical transport measurements at applied magnetic fields" I would suggest to modify this to "The upper critical field towards 0 K at 124 GPa is determined to be between 56 and 78 T by extrapolation of the results of the electrical transport measurements at applied magnetic fields".

Further comments

- Was the resistance measured on warming or on cooling? What temperature sweep rates were used? What electronics was used to measure resistance and to control temperature?
- p. 6 please add the reference to Baumgartner et al. for the WHH equation simplification.
- in my opinion, the data presentation with a color bar in Fig. 3a is not acceptable. Each of the experimental curves should be uniquely and unambiguously identified.
- Fig. 4a - please use counts (with numbers) for Y-axis.
- text and Table S1: how and with what accuracy the synthesis temperature was determined?
- Fig. S3A - please add an inset with the data in 0-20 K range. Is the feature due to superconductivity in unreacted Ce-Y alloy or an artifact of the measurements system.
- Fig. S3b - can the authors please show how T_c was determined for each curve. With such $R(T)$ shape it is not really clear.

On Response to Referee

- the authors write; "The realization of the zero-resistance state as well as the suppression of the superconducting transition upon the magnetic field clearly supports the bulk superconductivity in the studied Ce-Y-alloy hydrides. " This statement is incorrect. Only a percolation path is required to have zero resistance. Neither existence of this path, nor magnetic field dependence of the observed resistance, can support a statement of bulk superconductivity. Careful study of $R(T)$ measured at different currents might give some (still incomplete) support of such statement.
- the authors write "For most magnetic elements, the magnetic moment could be suppressed or quenched by the applied pressure." This is a very general statement that is not really helping their arguments. A clear issue is at which pressure would this happen for a particular rare earth (in this case Ce) in a particular structure / compound. A counter-example much closer to home would be the work <https://doi.org/10.1002/adma.202204038> on Nd impurities in LaH₁₀.

To summarize, I would not object publication of the revision of this text in Nature Communications, provided the comments above are addressed. This said, I consider this work a borderline for Nat. Commun.

Summary of changes on manuscript No. NCOMMS-22-29561B

We have adopted all the suggestions and comments from three reviewers and editor to revise the manuscript from every aspect. The detailed changes are summarized as follows:

To 1st Reviewer

- 1) We have changed the words “modest pressures” into “high pressures” in the title and other places of the main text of our manuscript.
- 2) We have replotted Fig. 2b to include our results reported in the present work with the comparison of the experimental data points of YH_9 and CeH_9 .

To 2nd Reviewer

- 1) We have removed the following expression in the revised manuscript.

“In the XRD patterns we failed to detect any peaks corresponding to ...elemental Y or Ce”.

- 2) We have replotted Fig. 4a with all the measured features of the collected diffraction data.

To 3rd Reviewer

- 1) We have revised the following description in the abstract:

“The superconductivity is obtained from the observed zero-resistance state with the detected onset critical temperatures in the range of 97-141 K.”

- 2) We have revised the following description in the abstract:

“The upper critical field towards 0 K at 124 GPa is determined to be between 56 and 78 T by extrapolation of the results of the electrical transport measurements at applied magnetic fields.”

- 3) We have revised Fig. 3a with the magnetic field value for each curve.
- 4) We have replotted Fig. 4a by using counts with numbers for the y-axis.

- 5) For the temperature estimation during the sample synthesis, we estimated the temperature in the range of 1500-2000 K. The laser power of 5 mW was used to focus on the sample with the laser spot around 1.5 micrometer. We have added these descriptions in the revised manuscript.
- 6) We have replotted Supplementary Fig. 3a by adding an inset with the data in the temperature range in 0-20 K.
- 7) We have replotted Supplementary Fig. 3b by adding the arrows for the onset superconducting transitions.
- 8) We have removed the GL equation and the WHH equation in the text by just citing the references because these are well known.

Responses to the reviewers' comments in their third reports on manuscript No. NCOMMS-22-29561B

We thank the three reviewers for the comments to the third version of our manuscript. Now we would like to provide a *point-by-point* response to the reviewers' comments as follows:

To Reviewer #1

Comment: "This is the 3rd review report for manuscript 380516 entitled 'Synthesis and superconductivity in yttrium-cerium hydrides at moderate pressures.'"

"After two previous reviews, the authors revised the manuscript to address the criticisms and questions of three reviewers. For this reviewer's comments, the authors' addresses are mostly satisfactory."

Reply: We feel happy to learn that the 1st reviewer satisfies to our responses to his/her previous comments and/or suggestions.

Comment: "This reviewer's main concern was the misleading writing that said the Y-Ce-hydride superconductor is synthesized and stabilized at moderate pressures. In the current manuscript, that claim was dropped, and the authors focus their discussion on the Tc and structural phases of the synthesized Y-Ce-hydrides, not comparing the stability pressure region with Y-hydrides and Ce-hydrides."

"Regarding the title of the manuscript, as stated by the author, this reviewer thinks it is better to change the words 'modest pressures' -> 'high pressures' per the discussion of the manuscript."

Reply: Fully adopted the reviewer's suggestion, we have changed the words "modest pressures" into "high pressures" in the title and other places of the main text.

Comment: "Besides, this reviewer suggests including the figure 'Tc vs. Pressure' for CeH_x, YH_x, and Ce-Y-H_x shown in response to this reviewer's comments. This figure certainly helps readers understand the achievement of this study much better than comparing the data on tables or texts."

Reply: We accept this nice suggestion from the reviewer and update our early version of Fig. 2b. The new figure includes the comparison of our results for Y_{0.5}Ce_{0.5}H₉ presented in the current work with the experimental data points of YH₉ from two groups and CeH₉ from two groups (one from us) as well. These compounds share the same crystal structure at the studied pressure ranges.

Comment: "As a summary, this reviewer thinks the current manuscript can be recommender for publication in Nature Communication. Further review is not necessary for this reviewer."

Reply: We would like to express our thanks to the reviewer again for this recommendation of this work for publication in *Nature Communications* after the improvements of every aspect in the response to his/her professional review, suggestions, and comments.

To Reviewer #2

Comment: “Referee 2 Suggestion II: The authors say they `want to *make the effort* to provide such information by Rietveld analysis’. However, my point was that Rietveld analysis is not appropriate, because the measured relative diffraction intensities are not sufficiently well sampled. It doesn’t matter how much effort is expended if the information is not present in the data due to inadequate statistical sampling of crystallite orientations. The authors do not make any claims in relation to crystal structure that would rely on accurate intensities (e.g. atomic positional parameters), so the use of Rietveld is not only unjustified, but also unnecessary.”

Reply: We totally agree with the second reviewer on this point. The Rietveld analysis is not appropriate for refining the structural parameters based on the collected diffraction data for hydrides at such high pressures. To pin down the structural information, one will need to grow single crystals in diamond anvil cells with large opening for the single-crystal diffraction study. Meanwhile, the crystal should be measured with the evidence for supporting superconductivity. So far, there are a lot of technique challenges in doing so. Now, the lattice parameters are obtained by fitting the diffraction peaks based on the Le Bail method. We have revised Fig. 3a for such considerations. The structural analysis has been updated for such changes in the main text and supplementary information of the revision of the manuscript.

Comment: “Referee 2 Suggestion III: the authors claim there (their) diffraction data confirm stoichiometry because `In the XRD patterns we failed to detect any peaks corresponding to ...elemental Y or Ce’. However, there were unindexed and undiscussed peaks in the 2d pattern they show, which I highlighted in my suggestion IV. How is it possible to know that these peaks do not come from an unidentified phase containing Y or Ce?”

Reply: Indeed, the diffraction data cannot be used to rule out the possible occurrence of elemental Y or Ce. In the present study, the starting alloy is $Y_{0.5}Ce_{0.5}$. If the hydrides based on it are formed, we assume the compounds in the formula of $Y_{0.5}Ce_{0.5}H_x$. By comparing the equations of states of the elements and hydrogen, we can determine the hydrogen level as 9. Our theoretical calculations support the lattice and dynamic stability for the phase of $Y_{0.5}Ce_{0.5}H_9$. The Raman spectra exhibit the phonon peaks as indicated from the calculated phonon dispersion. The experimentally obtained T_c values are in fair agreement with the calculations.

On the other hand, if Y or Ce is precipitated from the designed alloy in the experiments, we would detect superconductivity of either elemental Y or Ce at the studied pressures with comparable T_c values as reported previously. Many of our samples do not exhibit superconductivity corresponding to the case reported for elemental Y or Ce. We thus safely assume that the observed superconductivity comes from Y-Ce-H rather than the element Y or Ce.

Based on these considerations, we think that the statement of the formed compounds including $Y_{0.5}Ce_{0.5}$ are physically plausible. We wish the reviewer would agree with us for this point. We have revised the manuscript with the removal of the description of “In the XRD patterns we failed to detect any peaks corresponding to ...elemental Y or Ce”.

Comment: *“Referee 2 Suggestion IV: The authors have used a background subtraction method that incorrectly removes sharp, although weak, diffraction peaks, so these were not considered. This oversight allows for misinterpretation, such as the one I highlight above.”*

Reply: We accept the reviewer’s comment about the background subtraction. In the revision of the manuscript, we replotted Fig. 4a with all the features of the measured data.

Comment: *“In light of the comments above, I still feel there are areas of the manuscript where conclusions are drawn that exceed the limits of the data obtained.”*

Reply: We agree with the reviewer for the concern of the structural determination of the synthesized samples. As we mentioned, the final solution of this structural issue will depend on the accurate single-crystal diffraction analysis based on the superconducting crystal(s). The present manuscript mainly reports the sample synthesis at the conditions of pressure and temperature. Raman scattering, X-ray diffraction, and resistance measurements at zero-field and magnetic fields are combined all together to characterize the samples with the phases, structures, vibrational and physical properties. The theoretical calculations are conducted to understand the experimental observations.

Although the structures of the synthesized samples have not been accurately determined, the manuscript itself provides a large amount of information for the structural, vibrational, electrical transport, and superconducting properties on a single sample. The readers now can have the knowledge of a new superconductor $Y_{0.5}Ce_{0.5}H_9$ with higher T_c than the maximum value of CeH_9 at lower optimal pressure than that of YH_9 (Fig. 2b). We would like to express our appreciations to the reviewer for his/her early recommendation of this manuscript for the publication in *Nature Communications* as well as the above new comments. Now that the structural analysis method has been corrected and the diffraction data has been reanalyzed and replotted by keeping all the features from the measurements, the concerns from the reviewer might be safely removed. The reviewer’s recommendation should be well founded.

To Reviewer #3

Comment: "I would like to divide my report into two parts:

- comments on the text and SI;

- comments on the Response on Referee Letter

In my opinion, several statements in the text are not formulated with a proper precision and therefore might sound misleading:"

Reply: We appreciate the reviewer very much for suggesting us to improve the presentation and statements in a proper precision. We have seriously taken this nice suggestion in revising our manuscript one more time. The new version of the manuscript includes all the suggested changes.

Comment: "- abstract: `The superconductivity is detected with the zero-resistance state at the critical temperature in the range of 97-141 K...` The temperatures cited are the temperatures of the onset of the superconducting transition. Zero resistance state was not achieved in half of the experiments, Zero resistance state in cells 3, 6, 7 in Fig. 2a is achieved between ~115 K and ~10 K."

Reply: Adopting the reviewer's comment, we have revised such a statement in the abstract of the revised manuscript as the following.

The superconductivity is obtained from the observed zero-resistance state with the detected onset critical temperatures in the range of 97-141 K.

Comment: "- abstract: `The upper critical field towards 0 K at 124 GPa is determined with the large value of 78 T by the electrical transport measurements at applied magnetic fields`. I would suggest to modify this to `The upper critical field towards 0 K at 124 GPa is determined to be between 56 and 78 T by extrapolation of the results of the electrical transport measurements at applied magnetic fields`."

Reply: We thank the reviewer for this nice suggestion. We have adopted this suggestion and revised the description as suggested in the revised manuscript.

Further comments

Comment: "- Was the resistance measured on warming or on cooling? What temperature sweep rates were used? What electronics was user (used) to measure resistance and to control temperature?"

Reply: The resistance was measured on the warming run with the temperature sweep rates less than 1K/min. For the Cell-1, Cell-3, Cell-4, and Cell-5, the temperature was controlled by the 335-temperature controller with a Pt resistance sensor attached to the diamond-anvil cell close to the samples. For Cell-6 and Cell-7, the resistance was measured by PPMS from Quantum Design with their built-in temperature control

system.

Comment: “- p. 6 please add the reference to Baumgartner et al. for the WHH equation simplification.”

Reply: The simplification for the WHH equation by Baumgartner *et al.* is not derived, as it is only a fit function. There is no meaningful relation between its parameters and the parameters used in the derivation of the implicit WHH equation. It is only a good approximation because it agrees quite well with the numerical solution of the WHH equation. We thus keep the previous fitting and analysis by using GL and WHH formula.

Comment: “- in my opinion, the data presentation with a color bar in Fig. 3a is not acceptable. Each of the experimental curve should be uniquely and unambiguously identified.”

Reply: Adopting the recommendation from the reviewer, we have provided the clear magnetic field for each curve in Fig. 3a.

Comment: “- Fig. 4a - please use counts (with numbers) for Y-axis.”

Reply: Adopting the recommendation from the reviewer, we have used counts with numbers for the y-axis of Fig. 4a.

Comment: “- text and Table S1: how and with what accuracy the synthesis temperature was determined?”

Reply: The synthesis temperature is only a rough temperature. It is mostly located between 1500 and 2000 K. An estimation of higher than 1500 K seems appropriate. The more accuracy for the used method is the usage of laser power on the sample with the focused spot. In our experiments, we used the laser power of 5 mW to focus on the sample with the laser spot around 1.5 micrometer. We have added these descriptions in the revised manuscript.

Comment: “- Fig. S3A - please add an inset with the data in 0-20 K range. Is the feature due to superconductivity in unreacted Ce-Y alloy or an artifact of the measurements system.”

Reply: Adopting the recommendation from the reviewer, we have added an inset with the data in the temperature range in 0-20 K in Supplementary Fig. 3a. The feature is a signature of superconductivity of unreacted Ce-Y before laser heating.

Comment: “- Fig. S3b - can the authors please show how T_c was determined for each curve. With such $R(T)$ shape it is not really clear.”

Reply: Adopting the recommendation from the reviewer, we have added the arrows for the onset superconducting transitions in Supplementary Fig. 3b.

On Response to Referee

Comment: “- the authors write; ‘The realization of the zero-resistance state as well as the suppression of the superconducting transition upon the magnetic field clearly supports the bulk superconductivity in the studied Ce-Y-alloy hydrides.’ This statement is incorrect. Only a percolation path is required to have zero resistance. Neither existence of this path, nor magnetic field dependence of the observed resistance, can support a statement of bulk superconductivity. Careful study of $R(T)$ measured at different current might give some (still incomplete) support of such statement.”

Reply: We agree with the reviewer for the character of bulk superconductivity. In the revision, we concentrate on the synthesis and characterization of Ce-Y-H system in terms of multiple experimental techniques with the combination of the theoretical calculations. For superconductivity, we pay more attention on the data analysis rather than the argument of the bulk behaviour. The resistance measurements reveal the realization of superconductivity in the synthesized samples. $\text{Ce}_{0.5}\text{Y}_{0.5}\text{H}_9$ is identified as the major phase from the structural and spectroscopic analysis. The theoretical calculations based on the crystal structure of this major phase yield the experimentally observed T_c value. All these observations validate the reported findings presented in the present work. Finally, we would like to mention, we have completed the temperature-dependent resistance measurements on CeH_9 at different currents in our recent work [Ref. 18].

Comment: “- the authors write ‘For most magnetic element, the magnetic moment could be suppressed or quenched by the applied pressure.’ This is a very general statement that is not really helping their arguments. A clear issue is at which pressure would this happen for a particular rare earth (in this case Ce) in a particular structure / compound. A counter-example much closer to home would be the work <https://doi.org/10.1002/adma.202204038> on Nd impurities in LaH_{10} .”

Reply: Adopting the reviewer’s suggestion, we have made the extensive literature search for the high-pressure behaviours of elemental Ce. At high pressures, Ce undergoes three structural transformations from the initial face-centered cubic phase to an orthorhombic phase at around 1 GPa [Phys. Rev. 76, 301 (1949)] and then a monoclinic phase at around 5 GPa [PRL 78, 3884 (1997)] before eventually entering a body-centered tetragonal phase starting at 12-15 GPa [PRB 70, 014104 (2004)] while keeping this phase to the studied highest pressure of 208 GPa [JAP 85, 2451 (1999)]. The unpaired 4f electrons in Ce can be pushed into the conduction band upon lattice compression. These 4f electrons in the high-pressure phases were found to behave in an itinerant manner [J. Phys.: Condens. Matter 30, 395601 (2018)]. Benefitted from this interesting feature, superconductivity at T_c ranging from 50 mK to 1.8 K was reported experimentally from the second high-pressure phase on [PRL 211, 250 (1968) and PRB 108, 094502 (2023)]. At the pressure range as high as 100 GPa in the current study, one expects Ce to maintain its itinerant character. The non-magnetic consideration in

the theoretical calculations might be physically plausible. We would like to thank the reviewer for suggesting us to examine the validation of such a consideration.

Comment: *“To summarize, I would not object publication of the revision of this text in Nature Communications, provided the comments above are addressed. This said, I consider this work a borderline for Nat. Commun.”*

Reply: We appreciate the reviewer very much for this encouragement. In the new version of our resubmission, we have addressed every aspect of the suggestions and/or comments from the reviewer on the main text, supplementary information, and response to his/her comments. We would like to express our sincere thanks again to the reviewer if he/she could satisfy the revision together with our response and recommend our manuscript for the publication in *Nature Communications*.

REVIEWERS' COMMENTS

Reviewer #2 (Remarks to the Author):

I'm satisfied with the latest edits to the manuscript and can recommend publication.

Reviewer #3 (Remarks to the Author):

The authors have addressed many of my comments to the previous version of the text. Yet I would like to request to have two minor corrections/additions. After those are done, the manuscript would be suitable for publication in Nature Communications.

- please delete the word "unambiguously" on p.6
- please elaborate what "standard four-probe method" is in the electrical transport measurements (p. 11) In non-DAC literature, standard four-probe method implies linear arrangement of the electrical contacts with two outer ones being the current contacts and two inner ones the voltage. Here (Fig. 1b) the configuration similar to the one for van der Pauw was used. Are the results shown indeed coming from using the van der Pauw method?